



# A Global Total Column Ozone Climate Data Record

Greg E. Bodeker[1,2], Jan Nitzbon[1,3], Jordis S. Tradowsky[1], Stefanie Kremser[1], Alexander Schwertheim[1], and Jared Lewis[1]

[1]Bodeker Scientific, 42 Russell Street, Alexandra, 9320, New Zealand
[2]School of Geography, Environment and Earth Sciences, Victoria University of Wellington, New Zealand
[3]Permafrost Research Section, Alfred Wegener Institute Helmholtz Centre for Polar and Marine Research, Potsdam, Germany

**Correspondence:** G. E. Bodeker
(greg@bodekerscientific.com)

**Abstract.** Total column ozone (TCO) data from multiple satellite-based instruments have been combined to create a single near-global daily time series of ozone fields at 1.25° longitude by 1° latitude spanning the period 31 October 1978 to 31 December 2016. Comparisons against TCO measurements from the ground-based Dobson and Brewer spectrophotometer networks are used to remove offsets and drifts between the ground-based measurements and a subset of the satellite-based

measurements. The corrected subset is then used as a basis for homogenising the remaining data sets. The construction of this database improves on earlier versions of the database maintained first by the National Institute of Water and Atmospheric Research (NIWA) and now by Bodeker Scientific (BS), referred to as the NIWA-BS TCO database. The intention is that the NIWA-BS TCO database serves as a climate data record for TCO and, to this end, the requirements for constructing climate data records, as detailed by GCOS (the Global Climate Observing System) have been followed as closely as possible.

This new version includes a wider range of satellite-based instruments, uses updated sources of satellite data, extends the period covered, uses improved statistical methods to model the difference fields when homogenising the data sets, and, perhaps most importantly, robustly tracks uncertainties from the source data sets through to the final climate data record which is now accompanied by associated uncertainty fields. Furthermore, a gap-free TCO database (referred to as the BS-filled TCO database) has been created and is documented in this paper. The utility of the NIWA-BS TCO database is demon-

strated through an analysis of ozone trends from November 1978 to December 2016. Both databases are freely available for non-commercial purposes: the doi for the NIWA-BS TCO database is 10.5281/zenodo.1346424 (Bodeker et al., 2018) and is available from https://zenodo.org/record/1346424. The doi for the BS-filled TCO database is 10.5281/zenodo.3908787 (Bodeker et al., 2020) and is available from https://zenodo.org/record/3908787. In addition, both data sets are available from http://www.bodekerscientific.com/data/total-column-ozone.

## 20 1 Introduction

Total column ozone (TCO) has been identified as one of 50 essential climate variables (ECVs) by GCOS (Global Climate Observing System; GCOS-138 (2010); Bojinski et al. (2014)). Climate data records of ECVs serve a variety of purposes, e.g. climate data records of TCO are required to (1) assess the impacts of changes in ozone on radiative forcing of the climate





system, (2) assess the effectiveness of the Montreal Protocol for the protection of the ozone layer, and (3) determine the contribution of ozone changes to observed long-term trends in surface UV radiation. This paper presents an update of a database which has been used in many previous studies (e.g. Bodeker et al., 2001a, b, 2005; Müller et al., 2008). The database was first developed by NIWA (the New Zealand National Institute of Water and Atmospheric Research) and, in the last decade, has been

maintained and updated by Bodeker Scientific (BS). The non-filled database is hereafter referred to as the NIWA-BS TCO database and the filled database is referred to as BS-filled TCO database. The version 3.4 (V3.4) database reported on here extends from 31 October 1978 to 31 December 2016. In constructing this database, the guidelines for generating climate data records of ECVs detailed in GCOS-143 (2010) have been adhered to.

Improvements over earlier version of the database implemented in V3.4 include:

– New and updated sources of satellite-based TCO measurements are used, viz. data from NPP-OMPS (National Polar-orbiting Partnership-Ozone Mapping and Profiler Suite), GOME-2 (Global Ozone Monitoring Experiment-2) and SCIA-MACHY (Scanning Imaging Absorption Spectrometer for Atmospheric CHartographY) are now included in the combined data set. Various updates to previously used data sets are detailed in Section 2.

– Improved statistical methods are used to model the difference fields between data sets; zonal means of the difference
fields are modelled using Legendre expansions which comprise the meridional component of a spherical harmonic expansion which is best suited for statistically describing a field on a sphere (see Section 3 for more information).

– Measurement uncertainties on the source data sets, and the corrections applied to those data sets, have been collated and are propagated through to the combined ozone data set so that, for the first time, this data set is now provided with uncertainty estimates for each datum.

– Furthermore, the gap-free BS-filled TCO database has been generated (see Section 9) using a machine-learning (ML) algorithm that is trained to capture the broad-scale morphology of the TCO field which extends to regions where measurements are missing. The ML algorithm is based on regression of available data against NCEP (National Centers for Environmental Prediction) CFSR (Climate Forecast System Reanalysis) reanalysis tropopause height fields and against potential vorticity (PV) fields on the 550 K surface.

## 2  Source data

The various satellite-based TCO data sets used to create the version 3.4 NIWA-BS TCO database are summarized in Table 1. The time periods covered by the satellite data sets are shown graphically in Fig. 1. In addition to the information presented in Table 1:

– The four TOMS (Total Ozone Mapping Spectrometer) data sets (Adeos, Earth Probe, Meteor-3 and Nimbus-7) all use
the TOMS retrieval algorithm with Adeos using the version 7 algorithm and the remaining three using the version 8 algorithm.

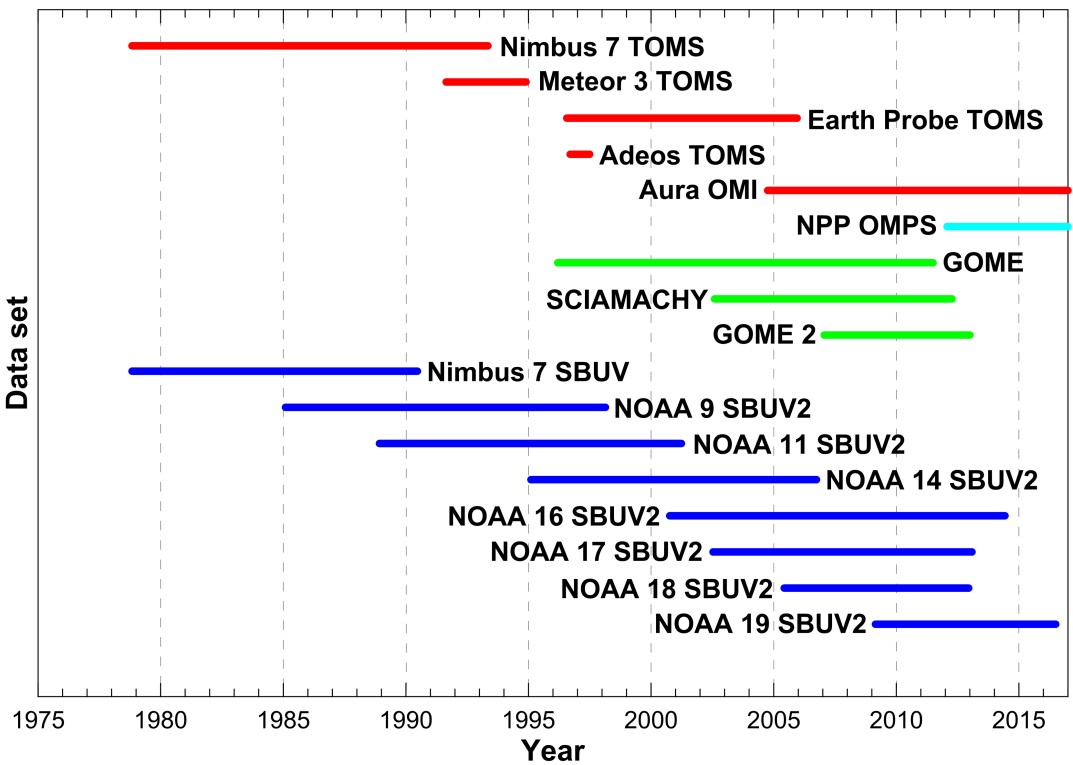

**Figure 1.** A graphical representation of the different satellite-based data sets used in this study and their periods of coverage.

- An algorithm similar to that used in the version 8 TOMS retrieval is used to conduct the version 8.6 retrievals of the SBUV (Solar Backscatter UltraViolet instrument) data (McPeters et al., 2013). Sparse gridded files were generated from the SBUV TCO measurements made at discrete locations as input to the process that creates the combined database.

- The high resolution and low resolution OMI (Ozone Monitoring Instrument) data sets both use a TOMS-like (version 8) retrieval algorithm.

- The GOME, GOME2 and SCIAMACHY data sets all use the GODFIT retrieval algorithm (Lerot et al., 2010).

- As stated in Section 1.1 of the OMPS Algorithm Theoretical Basis Document (ATBD), the algorithm used for retrieving the OMPS TCO is adapted from the heritage TOMS version 7 retrieval.

## 3 Determining corrections to TOMS and OMI data

First, the corrections required to the TOMS and OMI data sets are determined by comparing the satellite-based measurements with TCO measurements made by the global Dobson spectrophotometer and Brewer spectrometer networks. While TOMS



| Data set | Version | Period | Resolution (lon x lat) | Source |
|---|---|---|---|---|
| **TOMS/OMI** | | | | |
| Adeos | 7 | 1996-1997 | $1.25° \times 1°$ | ftp://toms.gsfc.nasa.gov/pub/adeos/ |
| Earth Probe | 8 | 1996-2005 | $1.25° \times 1°$ | ftp://toms.gsfc.nasa.gov/pub/eptoms/ |
| Meteor 3 | 8 | 1991-1994 | $1.25° \times 1°$ | ftp://toms.gsfc.nasa.gov/pub/meteor3/ |
| Nimbus 7 | 8 | 1978-1993 | $1.25° \times 1°$ | ftp://toms.gsfc.nasa.gov/pub/nimbus7/ |
| Aura OMI, low resolution | 8 | 2004-2012 | $1° \times 1°$ | http://tinyurl.com/p5qn9yd or http://tinyurl.com/qaeec7j |
| Aura OMI, high resolution | 8 | from 2004 | $0.25° \times 0.25°$ | http://tinyurl.com/klwcfbn or http://tinyurl.com/mmccw8q |
| **SBUV/SBUV2** | | | | |
| Numbus 7 | 8.6 | 1978-1990 | overpass data | ftp://toms.gsfc.nasa.gov/pub/sbuv/ |
| NOAA 9 | 8.6 | 1985-1998 | overpass data | ftp://toms.gsfc.nasa.gov/pub/sbuv/ |
| NOAA 11 | 8.6 | 1988-2001 | overpass data | ftp://toms.gsfc.nasa.gov/pub/sbuv/ |
| NOAA 16 | 8.6 | 2000-2003 | overpass data | ftp://toms.gsfc.nasa.gov/pub/sbuv/ |
| NOAA 14 | 8.6 | 1995-2006 | overpass data | ftp://toms.gsfc.nasa.gov/pub/sbuv/ |
| NOAA 17 | 8.6 | 2002-2013 | overpass data | ftp://toms.gsfc.nasa.gov/pub/sbuv/ |
| NOAA 18 | 8.6 | 2005-2012 | overpass data | ftp://toms.gsfc.nasa.gov/pub/sbuv/ |
| NOAA 19 | 8.6 | 2009-2013 | overpass data | ftp://toms.gsfc.nasa.gov/pub/sbuv/ |
| **ESA** | | | | |
| GOME | 1.01 | 1996-2011 | $1° \times 1°$ | http://tinyurl.com/mw9wg6h |
| GOME2 | 1.00 | 2007-2012 | $1° \times 1°$ | http://tinyurl.com/q4ad575 |
| SCIAMACHY | 1.00 | 2002-2012 | $1° \times 1°$ | http://tinyurl.com/lyz9alm |
| **OTHER** | | | | |
| NPP OMPS | 1.0 | from 2012 | $1° \times 1°$ | http://tinyurl.com/jvshwta |

**Table 1.** The source data sets used to create version 3.4 of the NIWA-BS TCO database.

and OMI data are provided in gridded data files, the original overpass data provide a higher quality data set for comparison with the ground-based measurement networks. To this end, overpass data from the four TOMS instruments and from the OMI instrument were obtained from the GSFC (Goddard Space Flight Center) FTP server (ftp://toms.gsfc.nasa.gov). Dobson and Brewer measurements were obtained from the WOUDC (World Ozone and Ultraviolet Radiation Data Centre). Three-

5 hourly means of the overpass data and direct-sun Dobson and Brewer TCO measurements were calculated for all sites for which Dobson and Brewer data were available. Exclusion of some of the Dobson and Brewer data, as discussed in Bodeker et al. (2001b), was required. Differences between 3-hourly means of ground-based and TOMS or OMI measurements were calculated. The uncertainties on the differences were calculated as the root sum of the squares of the uncertainties on the ground-based and satellite-based measurements (see Section 5).





The differences between the two data sets (satellite-based and ground-based) can be described as an offset and a drift i.e.

$$\Delta(t) = \alpha + \beta t \tag{1}$$

where $t$ is the time and $\alpha$ and $\beta$ are fit coefficients, denoting the offset and drift respectively, to be determined through a regression model fit to the differences. Because the offset and drift between the two data sets is likely to depend on season and

location, the $\alpha$ and $\beta$ coefficients are expanded in a Fourier series to account for the seasonality (see e.g. Bodeker et al., 1998) and then further expanded in spherical harmonics to account for the latitudinal and longitudinal structure in the difference field. Based on theoretical expectations and past experience (Bodeker et al., 2001b) we assume that the differences do not depend on longitude. Under this assumption, the spherical harmonic expansions reduce to Legendre polynomials. The $\alpha$ coefficient then takes the form:

$$\alpha = \sum_{l=0}^{N_{L,\alpha}-1} L_l(\theta) \left( \alpha_{l0} + \sum_{f=1}^{N_{F,\alpha}} \alpha_{lf,\sin} \sin\left(2\pi f t\right) + \alpha_{lf,\cos} \cos\left(2\pi f t\right) \right) \tag{2}$$

where $L_l$ denotes the $l^{th}$ Legendre polynomial and $\theta$ is the co-latitude (90° - latitude). A similar expansion is made for $\beta$. The choice of $N_{L,\alpha}$, $N_{F,\alpha}$, $N_{L,\beta}$, and $N_{F,\beta}$ is somewhat arbitrary; the values need to be set sufficiently high to capture the seasonal and latitudinal structure in $\alpha$ and $\beta$ but not so high as to over-fit the data and thereby introduce unrealistic structure into the statistically modelled difference field. Visual inspection of a wide variety of different choices of $N_{L,\alpha}$, $N_{F,\alpha}$, $N_{L,\beta}$,

and $N_{F,\beta}$ led to choice of (4,4,3,0) for the statistical model of the TOMS and OMI difference fields against the Dobson and Brewer networks where the Dobson and Brewer network is sparse, and (8,4,3,0) for differences from all other satellite-based data sources which are compared against the more dense, corrected, TOMS/OMI data set (see below). The resultant statistically modelled difference field is a compromise between high accuracy and low complexity or, equivalently, a compromise between simulating only meaningful structure in the difference field and avoiding over-fitting.

To avoid anomalous behaviour in the fit, which typically occurs at high latitudes and in regions where there are no satellite/ground-based difference pairs (e.g. during the polar night), the region between 80° and the pole is populated with difference values of zero for one month either side of the winter solstice. An example of one such fit is shown in Fig. 2. The morphology of this Dobson/Brewer - Nimbus-7 TOMS difference field is similar to that shown in Fig. 2 of Bodeker et al. (2001b) but with smaller differences resulting from the use of Legendre expansions in latitude, which better accommodate hemispheric asymmetries,

rather than a truncated polynomial expansion used in the earlier study. Similar $\Delta(t,\theta)$ difference fields (not shown) were statistically modelled for Adeos TOMS, Earth Probe TOMS, Meteor-3 TOMS, and Aura-OMI. Corrected TCO measurements for each of these data sets were calculated as follows:

$$TCO_{corr}(t,\theta,\phi) = TCO_{uncorr}(t,\theta,\phi) + \Delta(t,\theta) \tag{3}$$

where $\phi$ is the longitude.



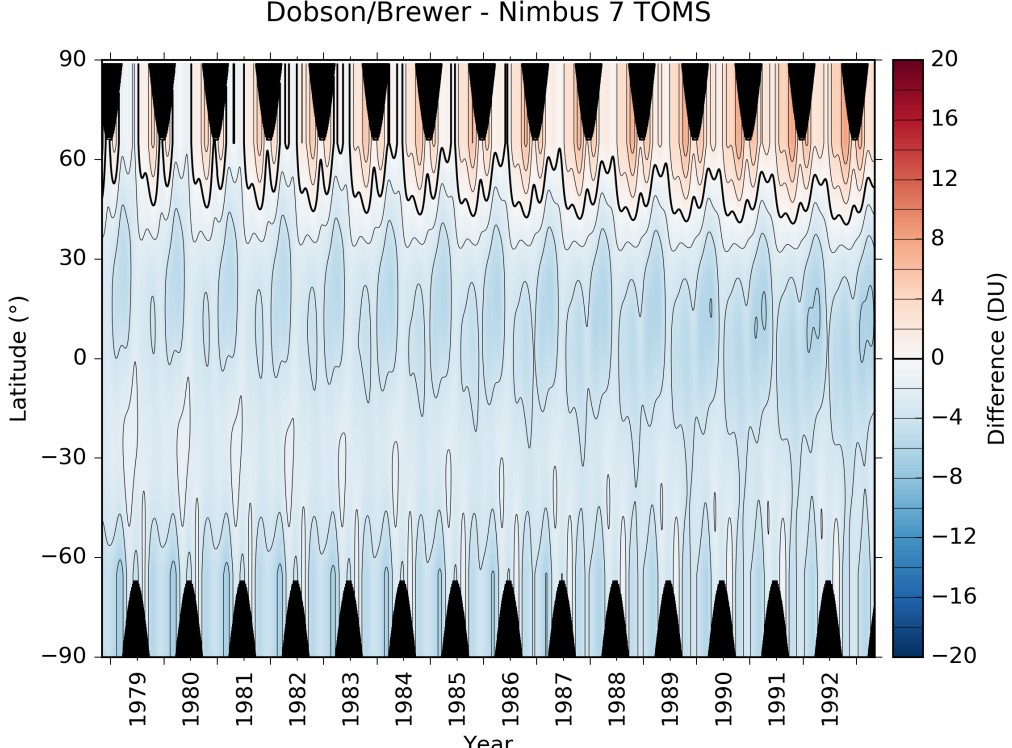

**Figure 2.** The results obtained by fitting Eq. 1 to the differences between Dobson/Brewer ground-based TCO measurements and Nimbus-7 TOMS overpass TCO measurements (ground-based minus satellite). Regions shaded in black denote the polar night where neither ground-based or space-based measurements are possible. The thick black line denotes the zero contour. Differences are shown in Dobson Units (DU; 1 DU = $2.69 \times 10^{16}$ molecules/cm$^2$).

## 4 Determining corrections to all other data sets

The TOMS and OMI grids, corrected for their offsets and drifts against the ground-based Dobson and Brewer measurements, now form the basis to correct the other data sets listed in Table 1. Differences between 1° zonal means from the combined corrected TOMS/OMI data and from the remaining data sets are calculated individually for each data set. The differences are then used as input to the regression model described in Eq. 1. If more than one TOMS/OMI meridional transect of zonal means is available for a given day, then all available difference values are passed to the regression model. As discussed in Bodeker et al. (2005), on 22 June 2003 a tape recorder failure on the ERS-2 satellite resulted in only a small portion of the Northern Hemisphere being sampled by the GOME instrument thereafter. To account for possible discontinuities in the difference field introduced by this anomaly, an additional basis function was included in the regression model for the TOMS/OMI-GOME



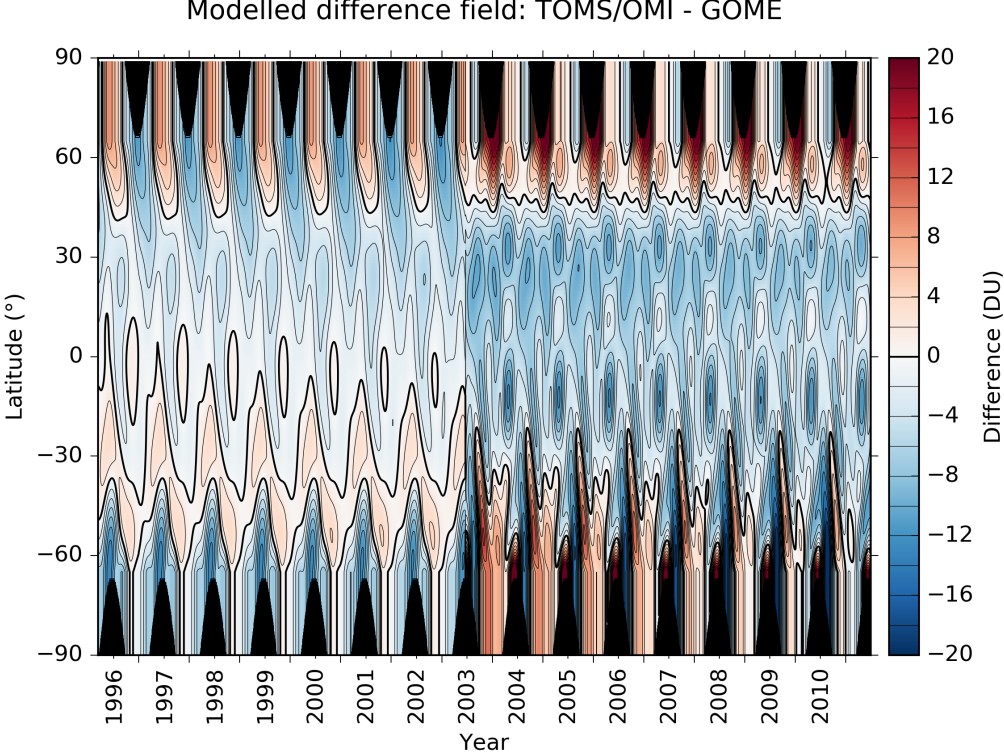

**Figure 3.** The results obtained by fitting Eq. 1 to the differences between corrected TOMS/OMI TCO zonal means and GOME TCO zonal means. An additional basis function is included to account for the 22 June 2003 anomaly. Regions shaded in grey denote the polar night where space-based measurements in the UV/visible part of the spectrum are not possible. The thick black line denotes the zero contour.

differences, set to zero prior to 22 June 2003 and to 1 thereafter. The resultant fit to the differences between zonal means of TOMS/OMI and GOME is shown in Fig. 3.

The effects of the 22 June 2003 anomaly are clear in Fig. 3 with higher and more variable differences after 22 June 2003 than before. Statistically modelled difference fields, similar to that shown in Fig. 3, were generated for all non-TOMS/OMI

5  data sets and extended, as required, to span the temporal coverage of each of those data sets to permit correction of the full data set. Because the combined TOMS/OMI record spans nearly the whole period (Nov 1978 to Dec 2016), extension into periods where TOMS/OMI data are not available is uncommon.

In addition to deriving the corrections for each data set listed in Table 1, the uncertainties on each of these corrections were also calculated since they contribute to the uncertainties of the respective data set as discussed in Section 5. The overall

10  uncertainties on each of the source data sets are used to create an uncertainty weighted mean of all source data sets to produce the final TCO databases (see Section 6).





| Data set | Random error TCO | Source | Link |
|---|---|---|---|
| Dobson | 1 % | Basher et al., 1980: Survey of WMO-sponsored Dobson Spectrophotometer intercomparisons | |
| Brewer | 1 % | Fioletov et al., 2005: The Brewer reference triad | |
| Adeos | 2 % | Krueger et al, 1998: ADEOS TOMS Data Products User's Guide | http://tinyurl.com/k4mdlpt |
| Earth Probe | 2 % | McPeters et al., 1998: Earth Probe TOMS Data Products User's Guide | http://tinyurl.com/ksrgprn |
| Meteor-3 | 3 % | Herman et al., 1996: Meteor-3 TOMS Data Products User's Guide | http://tinyurl.com/mm472kp |
| Nimbus-7 | 2 % | McPeters et al., 1996: Nimbus-7 TOMS Data Products User's Guid | macuv.gsfc.nasa.gov/doc/ n7usrguide.pdf |
| OMI | 2 % | OMI Algorithm Theoretical Basis Document Volume II, 2002 | http://tinyurl.com/knepwxk |
| all ESA | <1.7 % (SZA <80°) <2.6 % (SZA >80°) | GODFIT Algorithm Theoretical Basis Document, 2013 | http://tinyurl.com/lg27pzd |
| NPP OMPS | 1.12 DU, 0.64 % | OMPS NADIR TCO Algorithm Theoretical Basis Document, 2009 | http://tinyurl.com/mprd4zp |
| all SBUV | 5.0 DU | P.K. Bhartia: Personal communication, 2014. | |

**Table 2.** Uncertainties on the source data sets used in the construction of the TCO databases.

## 5   Uncertainties on the source data sets

One attribute of this version of the NIWA-BS TCO database that differentiates it from previous versions is the provision of
uncertainty estimates on each TCO value in the database. This development has been driven, in large part, by the requirements
for a climate data record as stipulated in GCOS-143 (2010). Table 2 gives an overview of the literature on which we have based
the uncertainty estimates of our source data sets. For the NPP-OMPS instrument, the uncertainty on each TCO measurement
comprises both a static component (in DU), and a component that scales with the TCO i.e. is a percentage of the TCO. The
relevant values (1.12 DU for the static component and 0.64 % for the component that scales with TCO) were derived from a
linear fit to the data listed in Table 7.3-7 of the Joint Polar Satellite System OMPS NADIR Total Column Ozone Algorithm
Theoretical Basis Document (2011).

The random uncertainties on the raw values listed in Table 2 are propagated through the analysis to result in an uncertainty
estimate on the final product. When regression modelling the difference field between the ground-based Dobson/Brewer mea-
surements and the TOMS/OMI overpass TCO measurements, the uncertainties passed to the regression model (Eq. 1) are:

$$\sigma_{\mathrm{diff}} = \sqrt{\sigma_{\mathrm{DB}}^2 + \sigma_{\mathrm{TOMS/OMIovp}}^2} \tag{4}$$

where $\sigma_{\mathrm{DB}}$ is the measurement uncertainty on the Dobson/Brewer measurements (1 %) and $\sigma_{\mathrm{TOMS/OMIovp}}$ is the measurement
uncertainty on the TOMS/OMI overpass measurements.

The uncertainty on the modelled difference field is calculated using a Monte Carlo approach whereby the uncertainties on
each difference pair are used to generate new estimates of the differences which then constitute a new data set of differences to





which the statistical model is fitted. The process is repeated 100 times. The mean and standard deviation of the 100 resultant model fits provides the final difference field and its uncertainty ($\sigma_\delta$). The uncertainties on the corrected TOMS/OMI values, calculated using Eq. 3, are then given by:

$$\sigma_{\text{Corr}}(\theta, \phi, t) = \sqrt{\sigma_{\text{uncorr}}(\theta, \phi, t)^2 + \sigma_\Delta(\theta, t)^2} \tag{5}$$

A similar procedure is used to propagate uncertainties in the corrections of the other satellite data sets against the corrected TOMS/OMI data sets. Recall that these corrections are based on comparisons of zonal means. To estimate the uncertainties on the zonal means, rather than taking the weighted mean of the single measurements, the unweighted arithmetic mean is calculated so that every measurement has the same weight. The zonal mean $ZM$, and its uncertainty $\sigma_{ZM}$, are then given by:

$$ZM = \frac{1}{N}\sum_i^N x_i \qquad\qquad \sigma_{ZM} = \frac{1}{N}\sqrt{\sum_i^N \sigma_i} \tag{6}$$

where $N$ is the number of measurements in the zone, $x_i$ are the measurements, and $\sigma_i$ are the uncertainties on the measurements. The uncertainty on the zonal mean also needs to account for the effects of any undersampling. If the zonal profile of TCO is highly structured, perhaps as a result of planetary-scale waves, then, if a particular space-based instrument does not fully capture that structure, the uncertainty on the zonal mean will be higher than it would have been the case otherwise. This is primarily a concern for the sparse sampling by the SBUV instruments used to create the combined database. For the

SBUV data sets, in addition to the zonal mean uncertainty calculated using Eq. 6, the potential uncertainty resulting from the sparse sampling was also accounted for. To estimate this additional uncertainty in the zonal means calculated using SBUV measurements an algorithm was developed to compare the zonal mean of a well-sampled zonal TCO profile with the zonal mean calculated using the same zonal profile but sampled at the SBUV measurement locations. The high spatial resolution (0.25°) OMI data set was used for this purpose. To estimate the potential zonal mean uncertainty for each day of the year

resulting from the SBUV under-sampling, grids of high spatial resolution data from OMI were considered for the 10 years 2004 to 2013 and for 10 days before and after the day of interest totaling 21 days. This gives a sample data set of 210 data grids. For each SBUV data set available on that calendar day, and for each latitude, two zonal means are calculated, viz (1) the true zonal mean ($ZM_{true}$) calculated from the 1440 values comprising the zonal TCO profile at 0.25° resolution, and (2) the sub-sampled zonal mean ($ZM_{sub}$) calculated using only those OMI data at the locations of the SBUV measurements. For each

latitude, 210 (from the 21 day window of 10 years of OMI data used) difference pairs of $ZM_{true} - ZM_{sub}$ can be calculated. If the SBUV sampling of the zonal mean was unbiased, all 210 values would be 0.0. The mean and standard deviation of these 210 values are then calculated and the standard deviation is used as an estimate of the SBUV sub-sampling uncertainty ($\sigma_{\text{subsample}}$) which is specific to a particular SBUV instrument and depends on the year and latitude. An example of one such sub-sampling uncertainty field for the NOAA 19 SBUV instrument is shown in Fig. 4.

The sub-sampling uncertainty maximizes during periods when the zonal profile shows more complex structure i.e. typically in winter and spring when mid-latitude planetary wave activity maximizes. The sub-sampling uncertainty on the zonal means



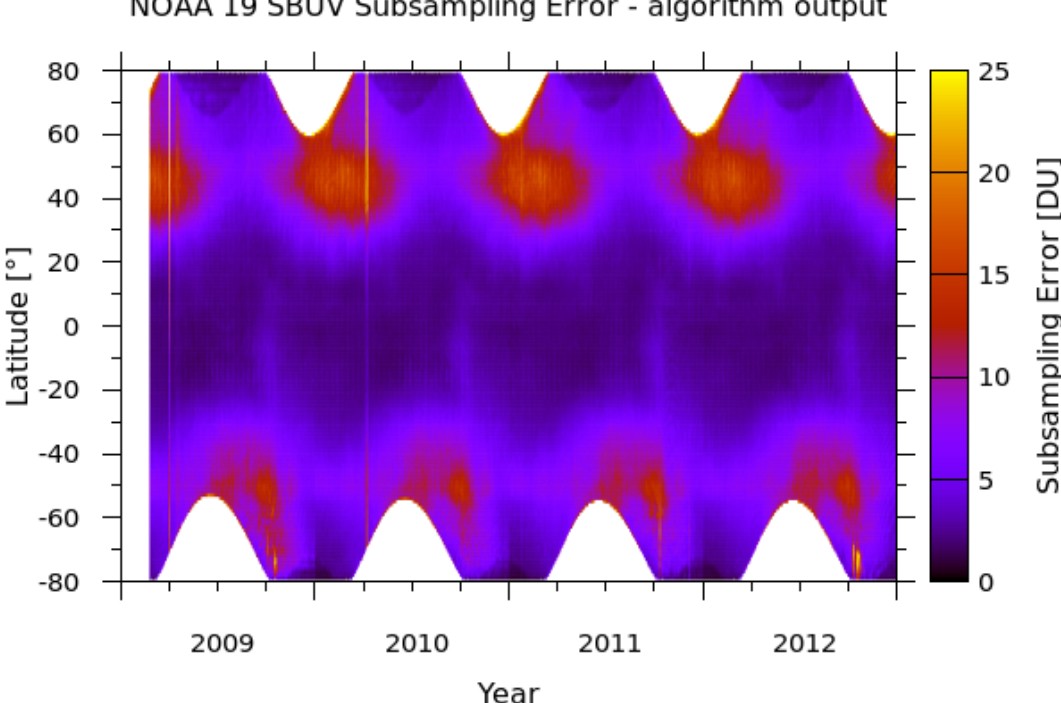

**Figure 4.** The additional uncertainty introduced to the SBUV zonal means as a result of under-sampling the zonal TCO profile. Regions shaded in white denote the polar night where space-based measurements in the UV/visible part of the spectrum are not possible.

is added to the zonal mean uncertainty calculated using Eq. 6 as:

$$\sigma_{\text{ZM}_{\text{SBUV}}} = \sqrt{\sigma_{\text{ZM}}^2 + \sigma_{\text{subsample}}^2} \tag{7}$$

## 6   Creating the combined data set

To construct a single TCO field for each day, a weighted mean of all available corrected measurements in each grid cell is

5   calculated. A grid of $1.25°$ longitude by $1.0°$ latitude was selected for the final product. The weights applied to the individual available TCO measurements are derived from the measurement uncertainties on each available measurement, viz:

$$\overline{TCO_{i,j}} = \frac{\sum_k w_{i,j,k} TCO_{i,j,k}}{\sum_k w_{i,j,k}} \qquad\qquad \sigma_{\overline{TCO_{i,j}}}^2 = \frac{\sum_k w_{i,j,k}^2 \sigma_{i,j,k}^2}{\left(\sum_k w_{i,j,k}\right)^2} \tag{8}$$

where $i$ and $j$ are indices over latitude and longitude, $k$ is an index over the measurements from different satellites in that cell, and the weights ($w_{i,j,k}$) are calculated as $1/\sigma^2$ where $\sigma$ is the measurement uncertainty incorporating any additional uncertainty

10   introduced by corrections made to the original data. Unlike previous versions of this TCO database, the daily combined TCO





| Data set | Instruments | Record length | Reference | URL |
|---|---|---|---|---|
| SBUV V8.6 NASA | BUV Nimbus-4, SBUV Nimbus-7, SBUV/2 NOAA 9 to 19 | 10/1978 to 7/2013 | McPeters et al. (2013); Frith et al. (2014) | http://tinyurl.com/m3m5qyn |
| SBUV V8.6 NOAA | SBUV Nimbus-7, SBUV/2 NOAA 9 to 19 | 11/1978 to 12/2012 | Wild et al. (2012); McPeters et al. (2013) | http://tinyurl.com/llbdssc |
| GSG Bremen | GOME, SCIAMACHY, GOME2 | 7/1995 to 7/2013 | Weber et al. (2013) | http://tinyurl.com/l3gz6w6 |
| ESA CCI | GOME, SCIAMACHY, GOME2 | 3/1996 to 6/2011 | Lerot et al. (2014) | http://tinyurl.com/q5h4bkj |

**Table 3.** Sources and details of the independent data sets used to validate the NIWA-BS total column ozone database.

fields are accompanied by fields of uncertainties and fields detailing the number of values that were averaged to produce the single combined value. An example of these three fields for one selected day is given in Fig. 5.

As expected, Fig. 5 shows that the uncertainty on TCO values decreases with an increasing number of source data sets. The regions of elevated uncertainty, sloping from north-west to south-east across the equator, arise from having only the OMI TCO
values available to build the mean. Cyan regions in panel (c) show where additional Earth Probe TOMS data contribute (with a resultant reduction in the uncertainty) and regions in green where SCIAMACHY additionally contributes data, reducing the uncertainties in the resultant mean to less than 3 DU.

## 7  Validating the combined data set

This new NIWA-BS TCO database has been validated through comparisons with the WOUDC database and four additional
independent TCO databases listed in Table 3. To account for different spatial resolutions, the NIWA-BS database was re-gridded to match the spatial resolution of each validation database. The differences reported in this section were calculated by subtracting the validation values from the NIWA-BS values such that positive differences represent elevated ozone values in the NIWA-BS database compared to the validation databases. Fig. 6 shows the globally averaged area-weighted differences between the NIWA-BS database and the validation databases over the full time period.
The NIWA-BS database displays a small negative bias (-0.2±2.7 DU) against the global mean monthly means calculated from the Dobson and Brewer measurements obtained from the WOUDC. A slightly larger negative bias (-1.2±1.2 DU) is seen in comparison with the SBUV V8.6 NASA time series. The bias against the SBUV V8.6 data set produced by NOAA slightly more negative but not statistically significantly different from zero (-1.3±1.5 DU). The comparison against the GSG Bremen

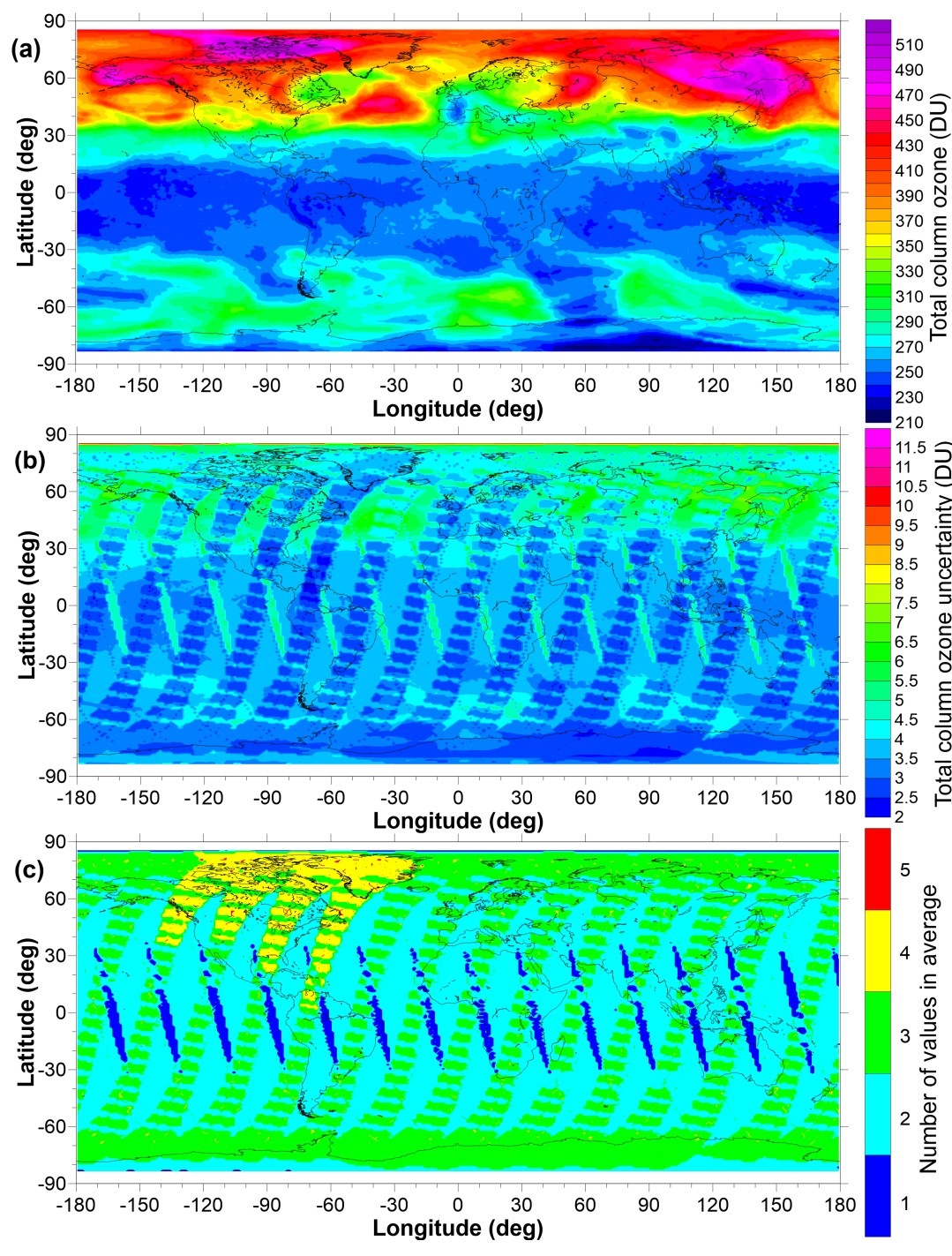

**Figure 5.** Example fields for 21 March 2005. (a) the TCO field, (b) the uncertainties on each value plotted in (a) and, (c) the number of values averaged to create the means plotted in panel (a). Regions shaded in white denote the polar night where space-based measurements in the UV/visible part of the spectrum are not possible.

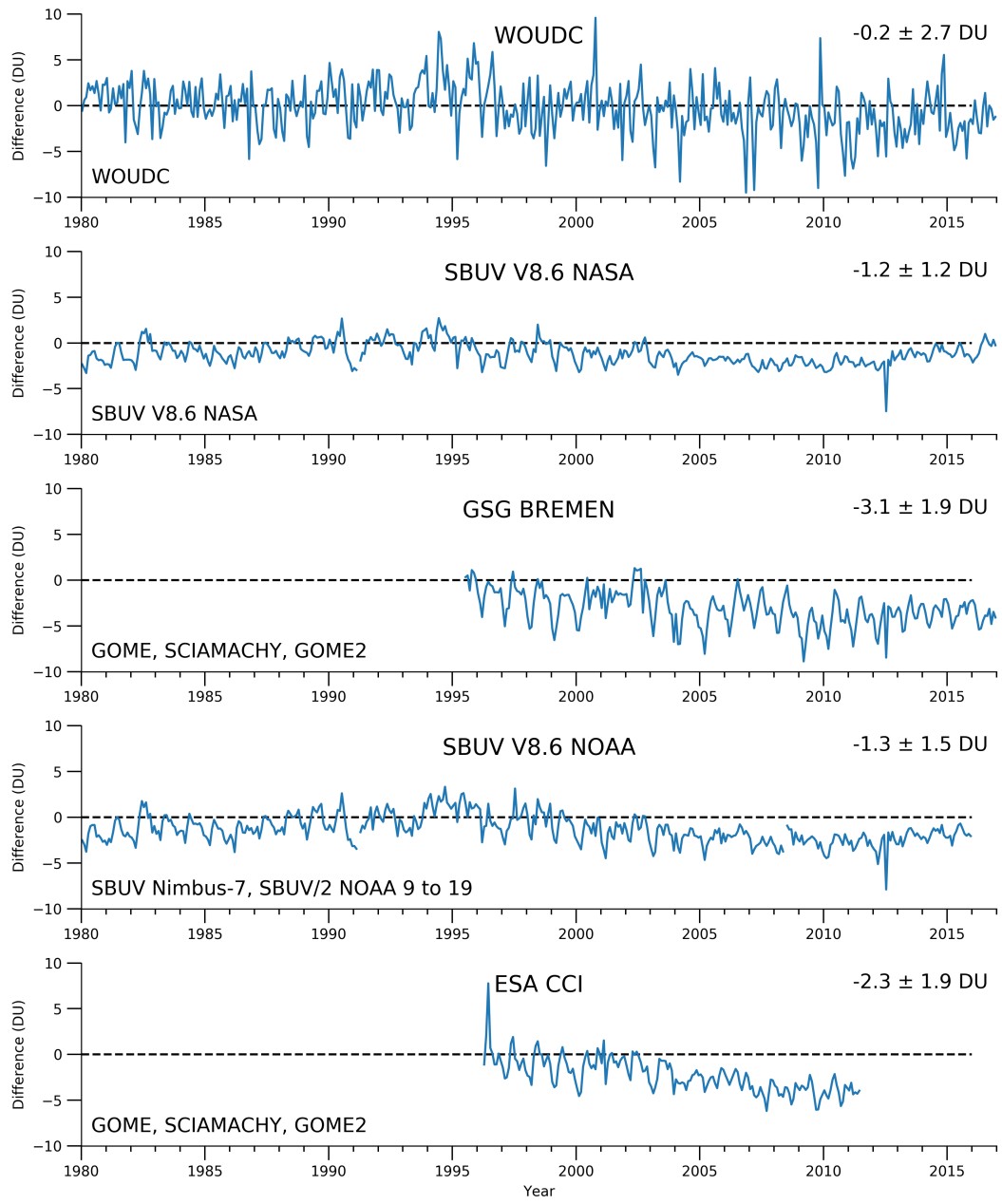

**Figure 6.** Area-weighted global mean monthly mean differences in TCO between the NIWA-BS database and the validation databases detailed in Table 3. The topmost panel shows the differences between the NIWA-BS database and the ground-based TCO database obtained from the WOUDC. The remaining four panels show differences against databases derived from space-based measurements. The statistics in the top right corner of each panel show the mean difference and standard deviation.

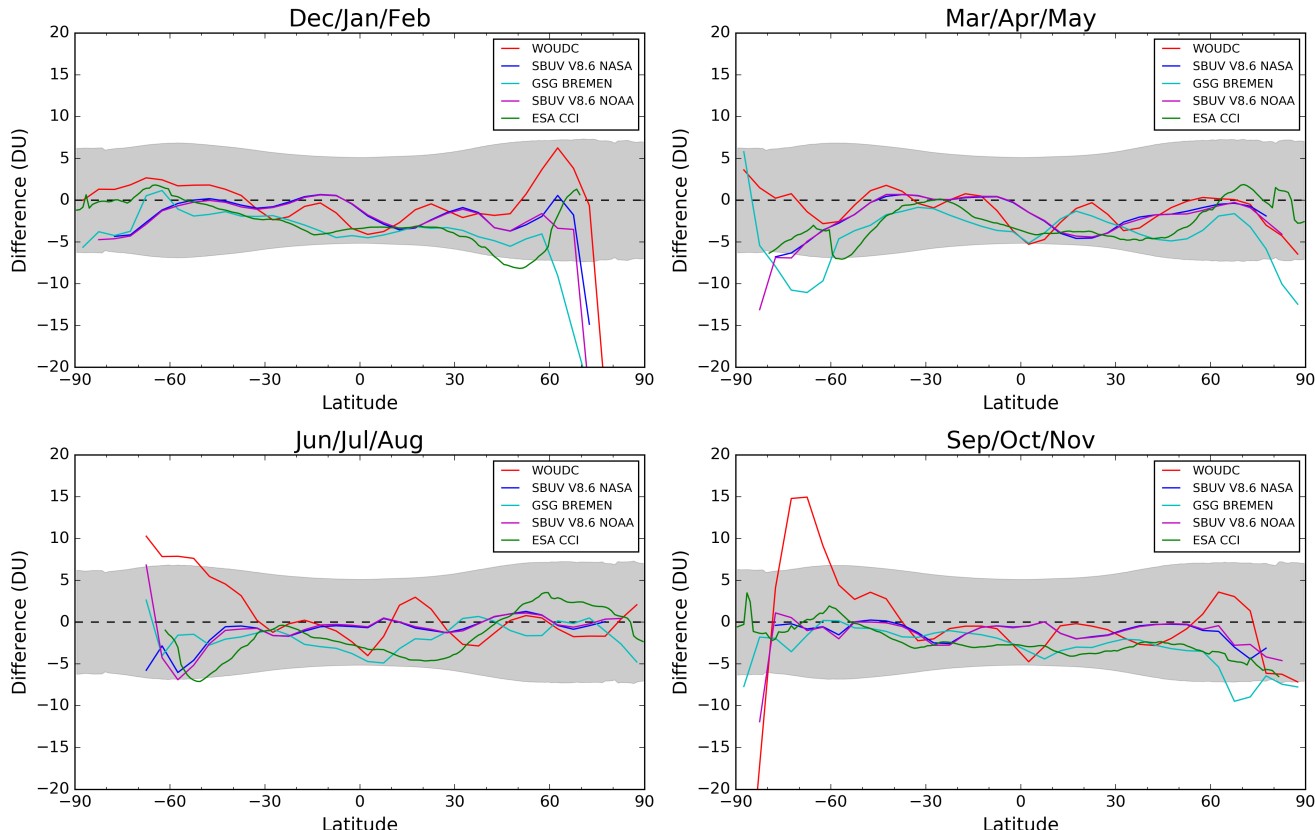

**Figure 7.** TCO differences (NIWA-BS TCO minus validation database) as a function of latitude plotted as seasonal means over the entire period of data available. The $1\sigma$ uncertainty range in the NIWA-BS TCO is shown in grey.

database suggests that the NIWA-BS time series exhibits a small anomalous downward trend starting around 2002 which is also reflected in the WOUDC comparison and in the ESA CCI comparison.

Seasonal mean differences between the NIWA-BS database and the five validation databases, as a function of latitude, are shown in Fig. 7. In general, the differences between the NIWA-BS TCO database and the validation databases are smaller than the uncertainties in the NIWA-BS database. This is not the case, however, in the high northern latitudes in winter where the NIWA-BS database shows statistically significantly smaller ozone values compared to the validation data sets. This results from larger differences in satellite measurements and ground-based measurements being inferred close to the region of permanent polar darkness where both satellite and ground-based measurements are scarce. A more in-depth comparison of the NIWA-BS database and the WOUDC database is presented in Fig. 8 where monthly mean zonal means (in 5° latitude zones) are differenced (NIWA-BS minus WOUDC). Over their full period of overlap, the mean difference between the data sets is -0.26 DU, with 95 % of the differences falling between -11.38 DU and 13.21 DU. Differences between the data sets are larger at higher





latitudes. A consistent feature of the NIWA-BS TCO database across most years is an overestimation in TCO equatorward of the Antarctic and an underestimation of TCO close to the South Pole with respect to WOUDC.

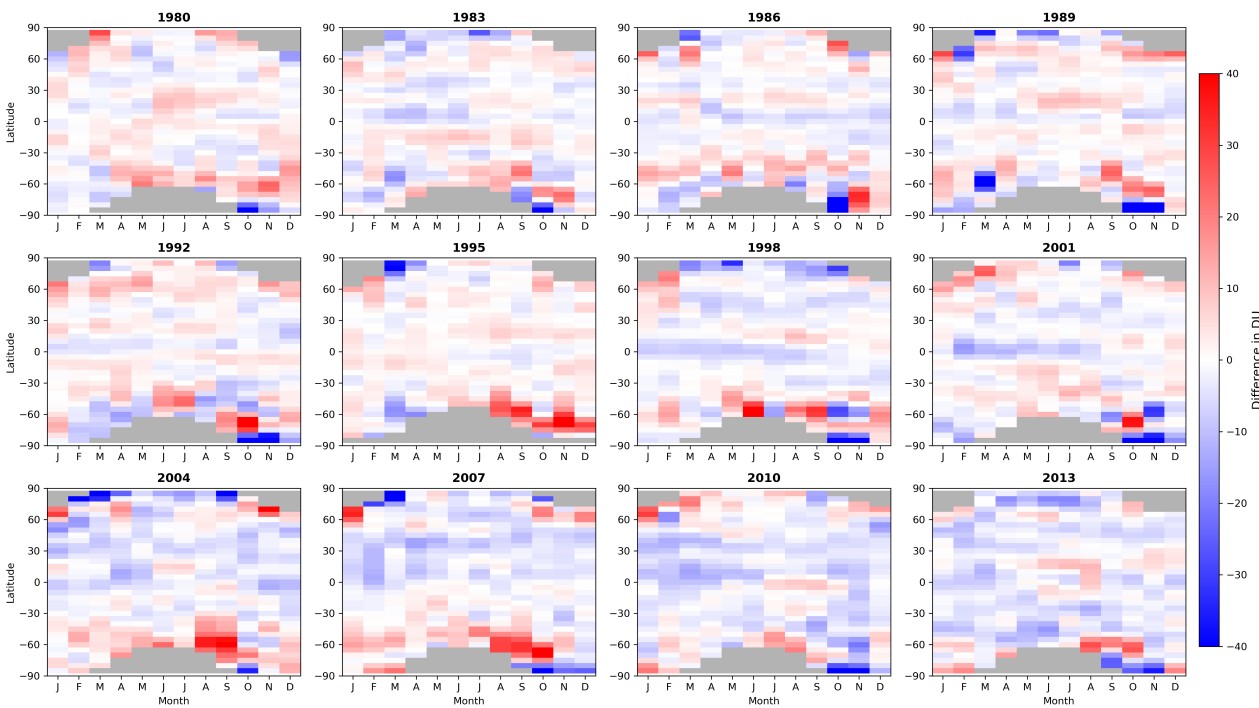

**Figure 8.** Monthly mean differences between NIWA-BS TCO and the WOUDC database for twelve selected years.

Differences between monthly mean 5° zonal means from the NIWA-BS database and the NASA SBUV merged ozone database Version 8.6 are shown in Fig. 9. The mean difference in TCO across the full overlap period is -1.21 DU, with 95 % of the differences in the range -5.96 DU to 2.95 DU. The differences are smaller in magnitude than those shown in Fig. 8 and show smaller year-to-year variability, perhaps as a result of the more dense spatio-temporal sampling by the SBUV instruments compared to the ground-based instruments. Features common across most years are an underestimation of TCO in the northern sub-tropics during the first half of each year and underestimations just equatorward of the polar night in the Southern Hemisphere.

As the NOAA SBUV database is available at daily resolution like the NIWA-BS TCO database, daily differences between zonal means from these two databases are calculated and shown in Fig. 10. Over their full overlap period, the average difference is -1.44 DU, with 95 % of the differences between -8.66 DU and 5.35 DU. Similar to the NASA SBUV comparisons, there appears to be a small underestimation in TCO over the northern sub-tropics in the first half of many years.

Differences between monthly mean 5° zonal mean TCO from the NIWA-BS database and the GSG Bremen TCO dataset which combined GOME, SCIAMACHY and GOME2 are shown in Fig. 11. There appears to be a consistent underestimation

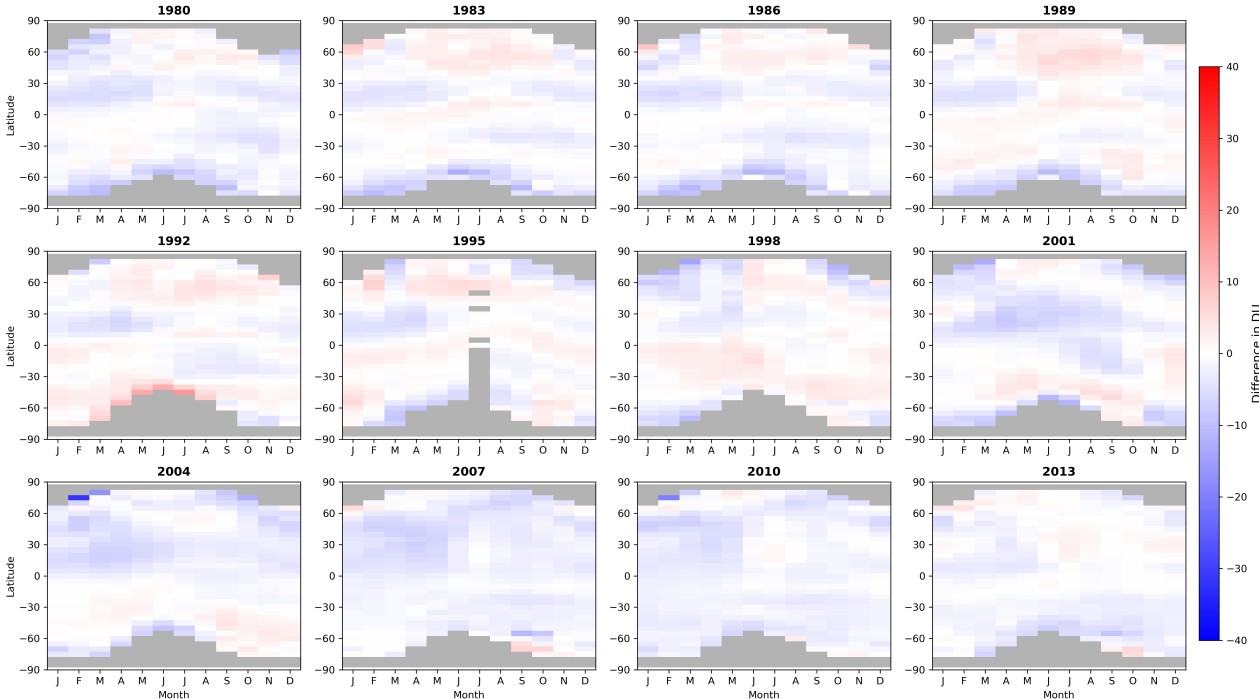

**Figure 9.** Differences between monthly mean zonal means calculated from the NIWA-BS TCO database and the NASA SBUV V8.6 database (NIWA-BS minus SBUV) for twelve selected years.

of TCO in the NIWA-BS database equatorward of the polar night from January to March in most years with respect to GSG Bremen. The mean difference between the databases is -3.14 DU, with 95 % of the differences lying between -9.79 DU and 2.81 DU.

Validation data from the ESA Climate Change Initiative (CCI) Level 3 TCO data set are available as monthly mean maps and differences between these monthly mean maps and NIWA-BS TCO are shown in Fig. 12. While there is significant spatial structure in some of the monthly difference fields, there is little structure that is consistent across multiple years. The mean difference is -2.36 DU, with 95 % of the differences lying between -10.63 DU and 6.34 DU.

## 8 Calculation of monthly mean and annual mean fields

Monthly mean TCO fields at $1.25°$ longitude and $1°$ latitude resolution (the same resolution as the daily fields) have been calculated, together with their uncertainties. The algorithm was used to calculate the mean and its uncertainty from $N$ measurements ($x_i$, i=1,...,$N$). The uncertainty on the mean is calculated in such a way that it depends on both the uncertainties on the measurements ($\sigma_i$) and on the variance in the measurements. First, a revised uncertainty for each datum is calculated to





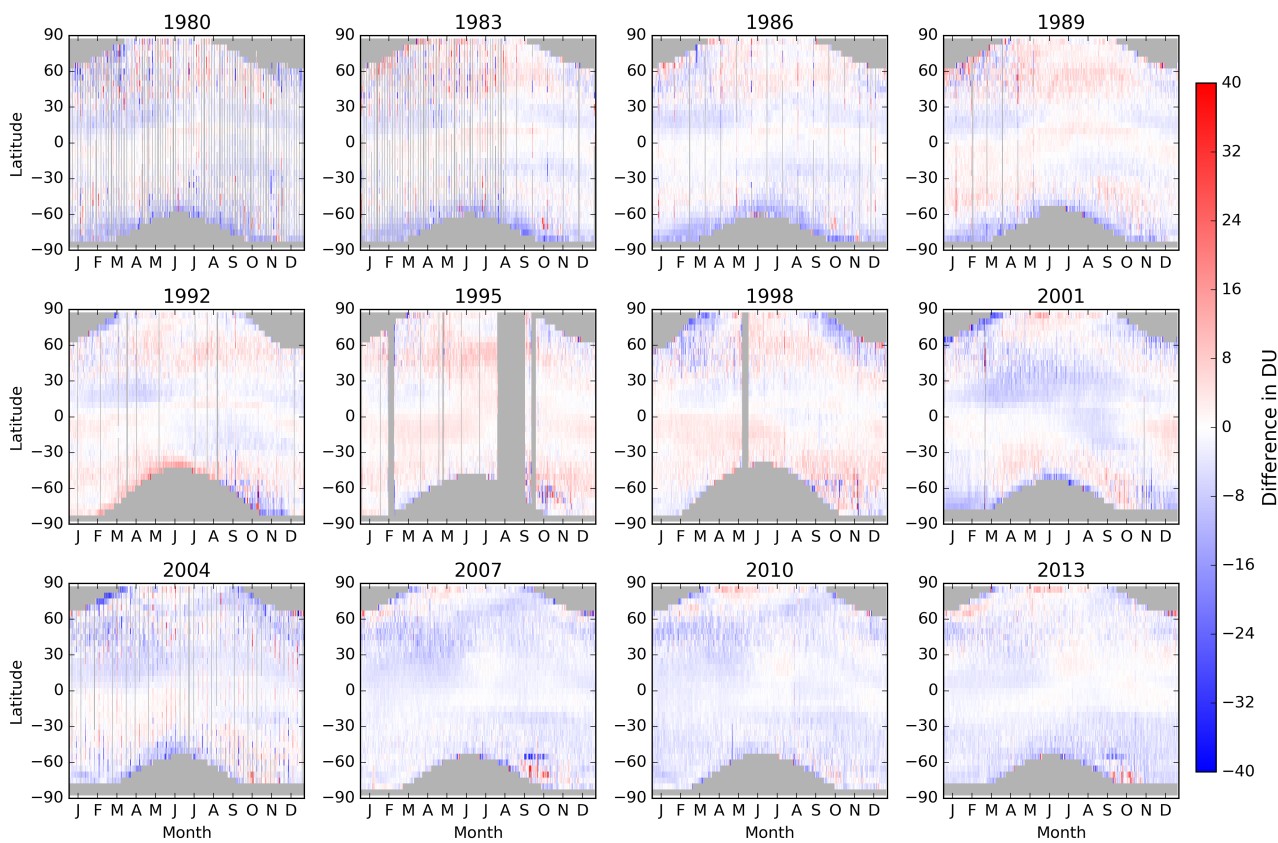

**Figure 10.** Daily differences between 5° zonal means calculated from the NIWA-BS TCO database and the NOAA SBUV V8.6 database for the same years as in Fig. 9.

reflect the true confidence we have on each measurement as an estimator of the mean:

$$\sigma_{i-new}^2 = \sigma_i^2 + (x_i - x_{i,exp})^2 \tag{9}$$

where $x_{i,exp}$ is the 'expectation' value which is taken to be the unweighted mean of the available measurements. The mean is then calculated as:

$$\bar{x} = \frac{\sum_{i=1}^{N} w_{i,new} \times x_i}{\sum_{i=1}^{N} w_{i,new}} \tag{10}$$

where $w_{i,new} = 1/\sigma_{i,new}^2$ and the uncertainty is calculated as:

$$\sigma_{\bar{x}} = \sqrt{\frac{\sum_{i=1}^{N} \sigma_{i,new}^2 \times w_i}{(NF - 1) \times \sum_{i=1}^{N} w_i}} \tag{11}$$



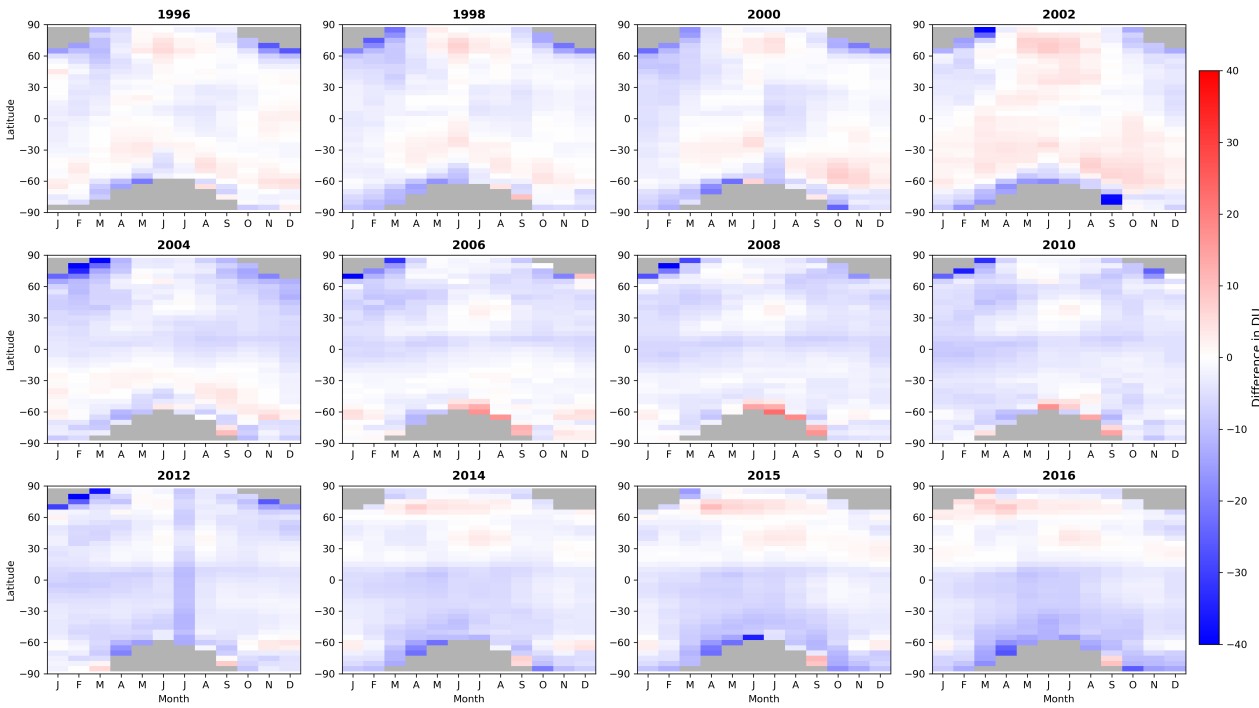

**Figure 11.** Differences between monthly mean $5°$ zonal mean TCO calculated from the NIWA-BS and GSG Bremen databases for twelve selected years (NIWA-BS minus GSG).

where $w_i = 1/\sigma_i^2$ and $NF$ is the degrees of freedom. In this case $NF$ was taken to be $N-1$ to account, in part, for auto-correlation in the daily time series used to calculate the monthly and annual means. The monthly mean and annual mean TCO fields are provided as a component of the version 3.4 database.

## 9   Creating the BS-filled Total Column Ozone database

5   For some applications, there is a need for gap-free TCO fields. To create a filled TCO database for a target day, the following steps are performed, each of which is detailed in sub-sections below.

1. A conservatively partially filled field is created (hereafter referred to as Field 1).

2. A machine-learning (ML) method is used to create a best estimate of the completely filled TCO field for the target day (hereafter referred to as Field 2).

10   3. The original unfilled TCO field, Field 1, and Field 2 are then 'blended' (using an algorithm described below) to generate the final filled field.



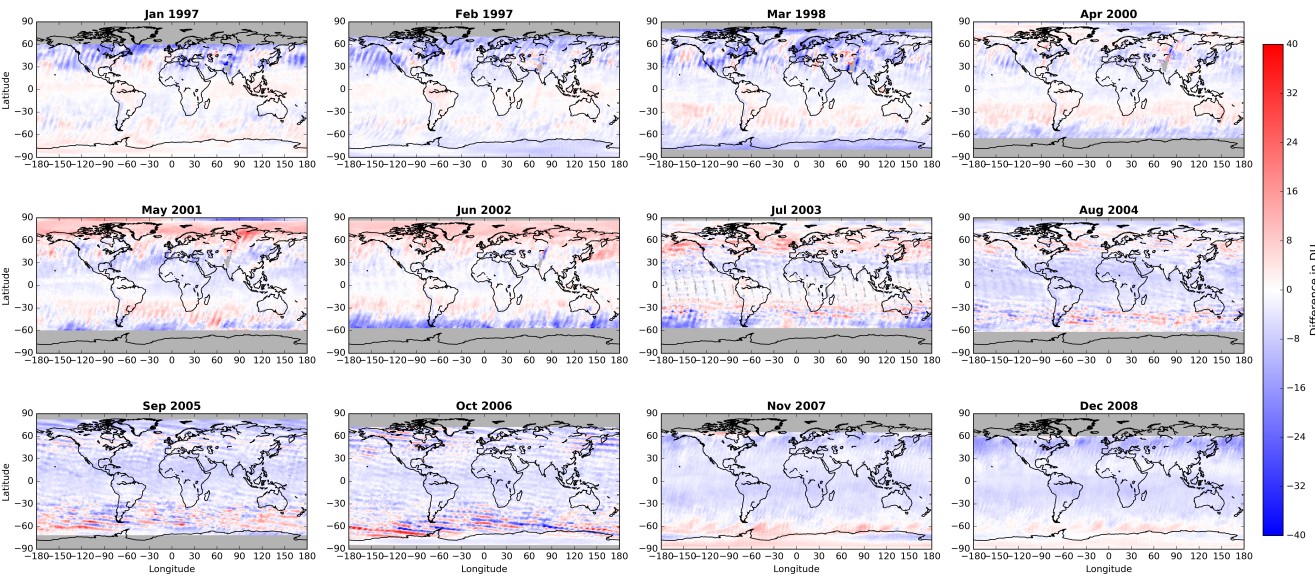

**Figure 12.** Differences between monthly mean TCO fields calculated from the NIWA-BS database and those available from the ESA CCI database, for twelve selected months/years.

The result is a TCO field that replicates the original data where they are available and, where no data are available, transitions preferentially into the conservatively filled field (Field 1). Where the conservatively filled field has missing data, we transition into the ML-filled field (Field 2). Each 'transition' is achieved by way of the blending process described below.

## 9.1 The conservatively filled field - Field 1

First a spatial nearest neighbour interpolation is used to fill as many missing values as possible. This is done by searching for cells with null TCO values that are neighboured, either to the north and south, or to the east and west, by non-null values. If such a non-null pair is found, that pair of TCO values, together with their uncertainties, is used to estimate the interstitial value which is taken as the mean of the two neighbouring values. Preference is given to zonal nearest neighbors. The uncertainty on the interpolated values is calculated by adding in quadrature the uncertainties on the neighbouring values.

After doing the spatial nearest neighbour interpolation, cases are sought where, for a cell containing a null value on day N, there are non-null values in the same cell on day N-1 and day N+1. Temporal nearest neighbour interpolation between the previous and following day is done in the same way as described for the spatial nearest neighbour interpolation.

Following the spatial-temporal nearest neighbour interpolation, a more extensive longitudinal interpolation finds two non-null values at the same latitude that are separated by two or more null values with the constraint that the non-null values cannot be separated by more than 30° in longitude. Linear interpolation between the two non-null values, including an estimate of



the uncertainties, is used to determine the interstitial values and their uncertainties. The spatial-temporal nearest neighbor interpolation, and the longitudinal interpolation, are repeated until no additional values are inserted into the grid. The result is a conservatively interpolated TCO field, still containing missing values, together with its original uncertainty field and traceable uncertainties on the newly interpolated values. A plot of the original TCO field, the conservatively filled field, and

the uncertainties on the conservatively filled field for day 3 of 1980 are shown in Fig. 13. The structure in the uncertainty field results from the propagation of uncertainties when calculating the conservatively filled field - larger uncertainties result when interpolated values are spatially far from available measurements.

## 9.2 The machine learning estimated field - Field 2

To create a completely filled TCO field for each day, a regression model, including an offset basis function, a tropopause height

basis function, and a potential vorticity at $550\,\mathrm{K}$ basis function, is trained on a 'window' of data around the target date. The trained regression model is used to generate a filled TCO field on the target date.

The regression model is of the form:

$$TCO_{i,j} = \alpha(\theta,\phi) + \beta(\theta,\phi) \times TH_{i,j} + \gamma(\theta,\phi) \times PV550_{i,j} + R_{i,j} \tag{12}$$

where $i,j$ subscripts denote indices over latitude ($\theta$) and longitude ($\phi$), $TH$ and $PV550$ are 6-hourly tropopause height fields

and $500\,\mathrm{K}$ potential vorticity fields, respectively, obtained from NCEP CFSR (National Centers for Environmental Prediction) CFSR (Climate Forecast System Reanalysis) reanalyses prior to 31 December 2010, and from NCEPCFSv2 reanalyses thereafter (Saha et al., 2010). $\alpha$, $\beta$ and $\gamma$ are fit coefficients and $R_{i,j}$ are the residuals that remain due to variance that cannot be explained by the regression model.

As denoted in Eq. 12, the three fit coefficients depend on latitude and longitude. That dependence is captured by expanding

the fit coefficients in spherical harmonics in a similar way as was done in Eq. 2, i.e.:

$$\alpha(\theta,\phi) = \sum_{l=0}^{N} \sum_{m=-l'}^{l'} \alpha_l^m Y_l^m(\theta,\phi) \tag{13}$$

where:

- $\theta$ is the co-latitude,

- $\phi$ is the longitude,

- $N$ is a regression model parameter,

- $l' \leq l$ where the exact limit for $l'$ is also a regression model parameter,

- $\alpha_l^m$ are the fit coefficients, and

- $Y_l^m(\theta,\phi)$ is the spherical harmonic function of degree $l$ and order $m$.

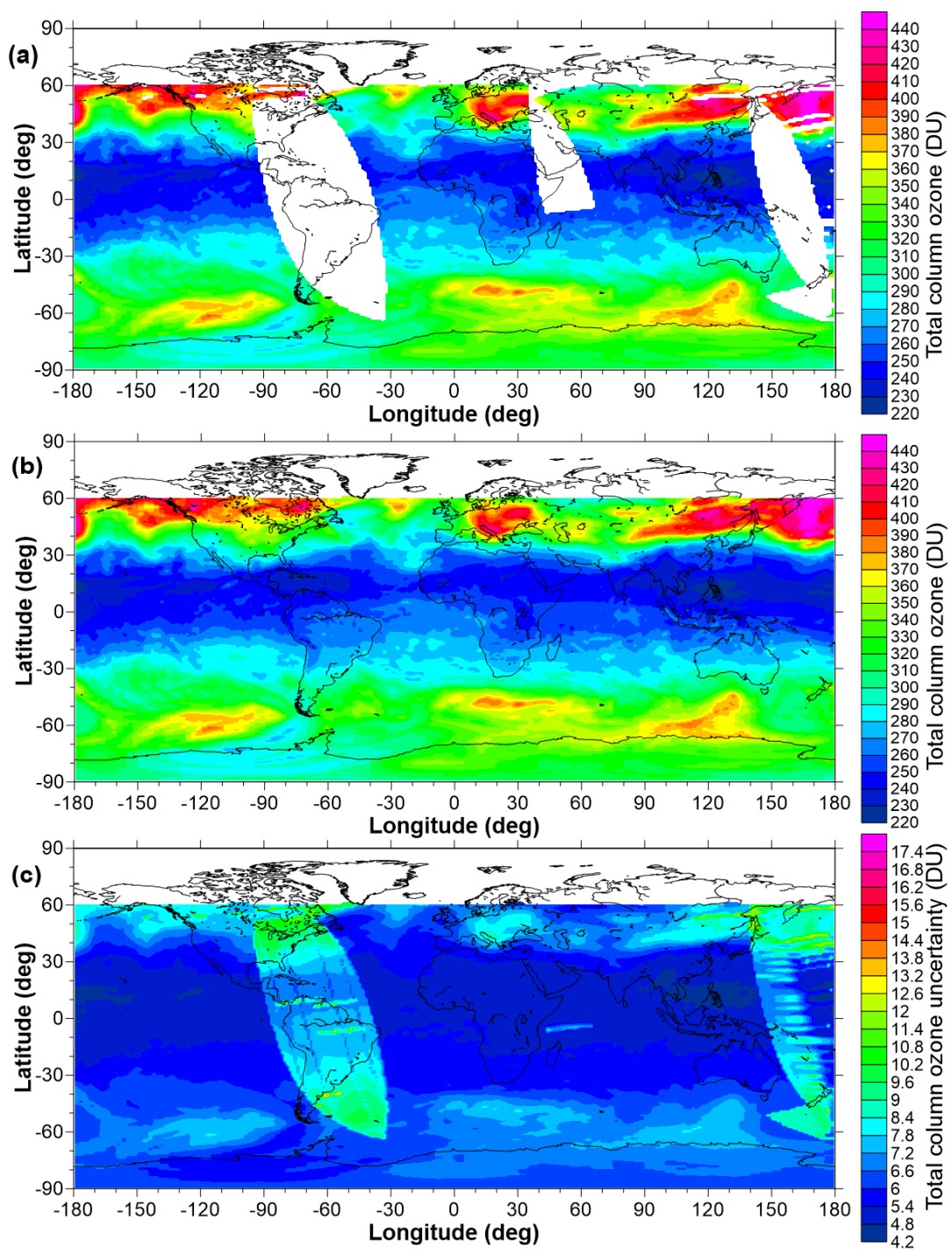

**Figure 13.** (a) The original unfilled TCO field on 3 January 1980, (b) the conservatively filled TCO field on the same day, and (c) the uncertainties on the conservatively filled field. White regions show where data are missing.



The $Y_l^m$ can be expressed as:

$$Y_l^m = \begin{cases} Y_l^0 & \text{if } m=0 \\ \sqrt{2}N_{(l,m)}P_l^m(\cos\theta)\cos(m \times \phi) & \text{if } m > 0 \\ \sqrt{2}N_{(l,m)}P_l^{-m}(\cos\theta)\sin(m \times \phi) & \text{if } m < 0 \end{cases} \tag{14}$$

For the purposes of fitting Eq. 14 to the TCO fields, the normalisation constants ($\sqrt{2}N_{(l,m)}$) can be ignored as they are taken up into the fit coefficients. When fitting Eq. 12, the initial $N$ value for $\alpha$ is set as 10 and for $\beta$ and $\gamma$ as 2. The maximum allowed

$N$ value for $\alpha$ is 10 and for $\beta$ and $\gamma$ is 5. The initial $l'$ value for $\alpha$ was set at 5 (with a maximum allowed value of 5) and for $\beta$ and $\gamma$ as 2 (with maximum allowed values of 5). These initial values and limits on the spherical harmonics expansions were selected based on careful consideration of the scales of spatial structures in TCO fields, and on how the dependence of TCO on $TH$ and $PV550K$ varies spatially. The training of the algorithm happens 'around' the target date as fields on neighbouring dates and years are used to establish the dependence of TCO on $TH$ and $PV550K$. A search ellipse, initially extending 3 days

either side of the target date, and 1 year either side of the target date, is defined to select TCO fields for the training where the ellipse is iteratively expanded until there are 20 TCO fields available for the training. The extension to neighbouring years is done because, in some cases, there are missing TCO fields in the current year such that a reliable fit of Eq. 12 cannot be performed. Within this search ellipse, the dependence of TCO on $TH$ and $PV550K$ at a similar time of the year is expected to hold.

From these 20 TCO fields, to avoid excessive computational expense, only up to 20,000 data points are passed to the regression model by sampling every $l^{th}$ value from all data available for training such that the total number of values passed is less than or equal to 20,000. The latitude and longitude of each ozone value is also passed to the regression model so that the associated $TH$ and $PV550K$ values can be extracted. The times associated with the TCO fields are assumed to be local noon times (since most of the satellites making the underlying measurements were sun-synchronous satellites with an equator-

crossing time close to solar noon). Therefore, the actual UTC time varies across the TCO field. The $TH$ and $PV550K$ values are linearly interpolated to those exact UTC times.

Various 'versions' of the regression model are tested i.e. different basis functions are excluded/included; the offset ($\alpha$) basis function is always included. In addition to switching different basis functions on/off, different values for $N$ and $l'$ are tested (perturbing these by $\pm 1$ around their start value) but ensuring that the maximum allowed zonal and meridional expansions

are not exceeded (see above). This results in many different possible constructs of the regression model. If any model, when evaluated over every latitude and longitude, results in a TCO value more than 10 % above the maximum measured TCO value passed to the regression model, or below 10 % less than the minimum value passed to the regression model, it is discarded to eliminate statistical models that significantly over-fit the TCO field. In addition to having a model with an excess of fit coefficients, over-fitting can also occur when anomalous values in the $TH$ or $PV550$ fields result in excessively high or low

TCO values being generated. This is why models that exclude/include these two basis functions are also tested. For all models that pass this initial test, a Bayesian Information Criterion (BIC; Liddle et al., 2007) score is calculated as:

$$BIC = M \times \ln(R^2/M) + NC \times \ln(M) \tag{15}$$





where $M$ is the number of data passed to the regression model, $R^2$ is a modified sum of the squares of the residuals, and $NC$ is the total number of coefficients in the fit. $R^2$ is modified to provide a strong disincentive for models generating values outside the range of measurements, i.e. where model values are below the minimum or above the maximum measurement passed to the regression model, the residual is inflated exponentially to impose an additional cost on the model for generating values outside

of the range of the data.

   Typically, for each daily TCO field, several hundred fits of the regression model are performed to find the optimal model construct (minimum $BIC$). This optimal model is then used to generate the statistically modelled TCO field. The database of different regression models is also used to calculate the structural uncertainties that result from different possible choices of spherical harmonic expansions. The uncertainties that result from uncertainties on the regression model fit coefficients are also

calculated. These two sources of uncertainty are then added in quadrature. The structural uncertainty statistics are calculated using only those regression modelled fields (out of the several hundred) that have the same sequence of basis functions switched on and off compared to the best fit. Examples of the unfilled TCO fields, the ML-modelled TCO fields and the uncertainty on the modelled TCO fields for days 304 and 305 of 1978 are shown in Fig. 14.

### 9.3   An algorithm for blending a primary and secondary TCO field

This section describes an algorithm to 'blend' some primary TCO field (hereafter Field A) with some secondary field (hereafter Field B) to create a single blended field (hereafter Field C) where the Field A values are preserved while smoothly transitioning to the Field B values. This algorithm is used below to combine the original TCO field and Field 1, and/or to combine Field 1 and Field2, and/or to combine the original TCO field and Field2; see Section 9.4.

   If there is a null value in a cell in Field A and a non-null value in the same cell in Field B then a proxy value for Field A is

found and combined with the value from Field B as follows:

$$C = W \times A_{proxy} + (1 - W) \times B_{value} \tag{16}$$

where $C$ is the blended value, $W$ is a weight calculated as detailed below, $A_{proxy}$ is a proxy value for the missing value in Field A determined as detailed below, and $B_{value}$ is the non-null value from Field B. To derive an $A_{proxy}$ value, 60° sectors around that missing grid node are scanned for non-null values from Field A which are within 20 grid cells in the east-west and

north-south directions. A weighted mean of the 6 values from those 6 search sectors is then calculated, where the weighting is calculated as:

$$W_i = cos \left( \frac{D \times \pi}{2 \times 10^6} \right) \tag{17}$$

where $D$ is the distance to the nearest non-null value in that sector measured in metres. The weight is set to zero when $D$ is greater than 1000 km. $A_{proxy}$ is then set to the weighted mean of the non-null values across all 6 sectors.

In calculating the blended value using Eq. 16, the weight ($W$) is calculated using the distance to the nearest non-null value across all 6 search sectors in Eq. 17. If no $A_{proxy}$ value can be found, then $W$ in Eq. 16 is set to zero. Standard error propagation

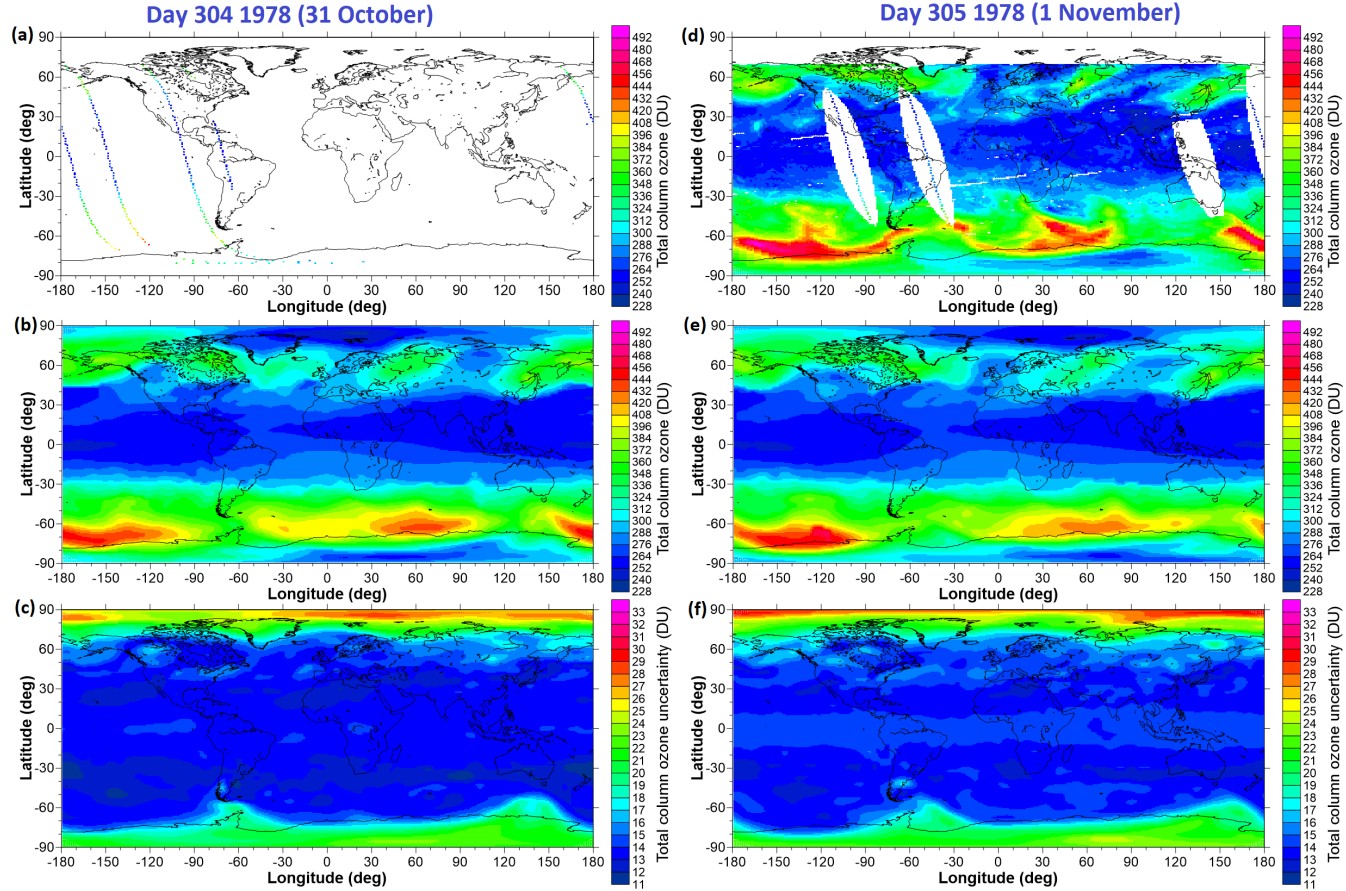

**Figure 14.** (a, d) The original unfilled TCO fields on 31 October 1978 and 1 November 1978, respectively; (b, e) the machine learning modelled fields; (c, f) the uncertainties associated with the machine learning modelled fields. On 31 October the optimal fit was obtained by expanding the offset basis function in spherical harmonics of degree 10 and order 5, expanding the tropopause height basis function in spherical harmonics of degree 4 and order 3, and the PV at 550 K basis function in spherical harmonics of degree 5 and order 3. For 1 November these expansions were (10,5) for the offset basis function (4,3) for the $TH$ basis function and (5,4) for the $PV550K$ basis function.





rules are used to determine an uncertainty on the blended value. This process results in using values from Field A where they
are available which then blend into values from Field B when they are not available.

### 9.4 Blending the original unfilled TCO field, Field 1, and Field 2 to construct the final filled field

For each day, the original TCO field, the conservatively filled field (Field 1) and the ML-modelled field (Field 2) are merged in
such a way that the original values are preserved where they are available. Where they are not available, the filled values relax
to the conservatively filled field where they are available. Where conservatively filled values are also not available, the filled
values relax to the ML-modelled field.

The ML-modelled fields, because they are modelled on $PV550K$ and $TH$ (which themselves can contain anomalous val-
ues), can occasionally display physically unrealistic spatial or temporal structures so, for any given day, to obtain a smoother
ML-filled field, a (1,4,6,4,1) weighting of the five daily ML-filled fields, centered on the day of interest, is calculated.

For the final blended data product, several possibilities exist for any given day, i.e.:

- *None of the three fields are available:* In this case no final filled field is generated.

- *Only the ML-modelled field is available:* In this case the ML-modelled field (possibly spatially smoothed), and its
  associated uncertainties, become the final filled field for the target day.

- *Only the conservatively filled field and the ML-modelled fields are available:* In this case the conservatively filled field
  (Field 1) and the ML-modelled field (Field 2) are blended to create the final filled field.

- *All three fields are available:* In this case the conservatively filled field and the ML-modelled field are blended to create
  an intermediate field. The original TCO field and the intermediate field are then blended to create the final filled TCO
  field and its uncertainty.

An example of the original field for TCO on 31 October 1978, the conservatively filled field, final field obtained from the
blending process, and the uncertainty on the final filled field are shown in Fig. 15 (the ML-modelled field for that day can be
seen in Fig. 14).

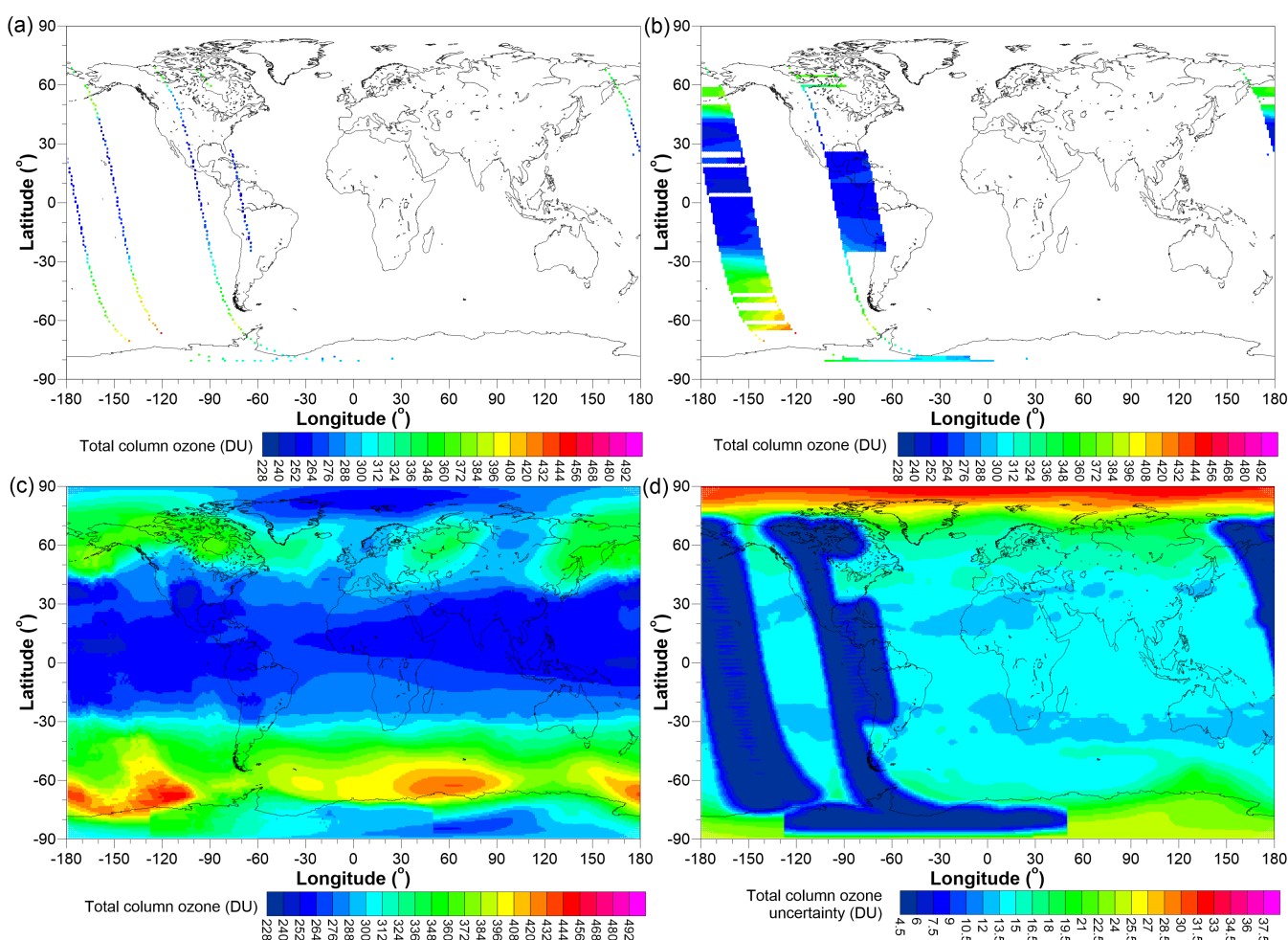

**Figure 15.** (a) The original unfilled TCO field on 31 October 1978, (b) the conservatively filled field, (c) the final filled field, and (d) the uncertainties on the final filled field.





## 10  Trend analysis

To prove the utility of the NIWA-BS TCO database, trends over the whole period have been diagnosed using the following regression model, described in detail in Bodeker et al. (2013):

$$
\begin{aligned}
\text{Ozone}(m,\theta,\phi) =\; & A(m,\theta,\phi)+ && \text{Max Fourier} = 4\\
& B(m,\theta,\phi) \times m/12+ && \text{Max Fourier} = 3\\
& C(m,\theta,\phi) \times m_{\text{m=0 if year<2000}}/12+ && \text{Max Fourier} = 3\\
& D(m,\theta,\phi) \times \text{QBO}(m)+ && \text{Max Fourier} = 2\\
& E(m,\theta,\phi) \times \text{QBO}_{orthog}(m)+ && \text{Max Fourier} = 2\\
& F(m,\theta,\phi) \times \text{ENSO}(m)+ && \text{Max Fourier} = 1\\
& G(m,\theta,\phi) \times \text{Solar}(m)+ && \text{Max Fourier} = 0\\
& H(m,\theta,\phi) \times \text{Pinatubo}(m+\Delta m)+ && \text{Max Fourier} = 1\\
& I(m,\theta,\phi) \times \text{El Chichón}(m+\Delta m)+ && \text{Max Fourier} = 1\\
& R(m,\theta,\phi))
\end{aligned}
\tag{18}
$$

where $\text{Ozone}(m,\theta,\phi)$ is the regression modelled TCO in month $m$ ($m$ = 1 to $NY \times 12$ where $NY$ is the total number of years of data) and at latitude $\theta$ and longitude $\phi$. The monthly mean TCO values were calculated as detailed in Eq. 10 and Eq. 11. Equation 18 is fitted independently to the monthly mean time series at each latitude and longitude. $A$ to $I$ are the regression model coefficients calculated using a standard least squares regression (Press et al., 1989).

The first term in the regression model ($A$ coefficient) represents a constant offset and, when expanded in a Fourier series, represents the mean annual cycle. In addition to the offset coefficient, each fit coefficient can depend on season, e.g., TCO trends vary with season. Therefore each coefficient is expanded in Fourier pairs as explained in Section 2.2 of Bodeker et al. (2015). The actual number of Fourier pairs for each regression coefficient is determined by finding the optimal set of expansions across all fit coefficients that minimizes a BIC as described above. The maximum number of Fourier pairs permitted for each regression model coefficient is listed in Eq. 18. The $B$ coefficients diagnose the trend over the full period while the $C$ coefficients diagnose the change in trend from 2000 onward. The QBO basis function was specified as the monthly mean 50 hPa Singapore zonal wind. The phase of the QBO varies with latitude and, to permit fitting of the phase, a second QBO basis function, mathematically orthogonalized to the first, was included in the regression model as was done in Bodeker et al. (2013).

The El Niño Southern Oscillation (ENSO), solar cycle, Mt. Pinatubo and El Chichón basis functions were the same as those used in Bodeker et al. (2001a). Note that the effects of the Pinatubo and El Chichón volcanic eruptions on TCO was delayed and so temporal offsets in those basis functions, denoted by the $\Delta m$, were permitted for up to two years where the $\Delta m$ values for each basis function were optimised as part of the same BIC optimisation used to determine the optimal Fourier expansions for



**Figure 16.** TCO trends for the period 1979 to 2000 evaluated at (a) day 90, (b) day 180, (c) day 270, and (d) day 360. The white areas at high latitudes in (b) and (d) result from there being insufficient data to establish meaningful trends during the polar night periods. The solid black line shows zero trend. Double hatched regions show where trends are not statistically different from zero at the $1\sigma$ confidence level, while single hatched regions show where the trend is not statistically different from zero at the $2\sigma$ confidence level.

each regression model fit coefficient. A one term auto-correlation model was used to account for the effects of auto-correlation in the residuals ($R(m,\theta,\phi)$) when calculating the uncertainties on the fit coefficients.

The $B$ coefficients, evaluated at day 90, 180, 270 and 360 through the year (noting that because these are Fourier expansions they can be evaluated at non-integral month values) are shown in Fig. 16. The spatial and seasonal pattern of ozone trends prior to 2000 is similar to numerous previous studies that have shown similar trend results. The $C$ coefficients, evaluated on the same days as the $B$ coefficients, are shown in Fig. 17. Note that Fig. 17 shows the **change** in trend compared to the trend shown in Fig. 16 i.e. areas midway between blue (more negative trends than before 2000) and red (more positive trends

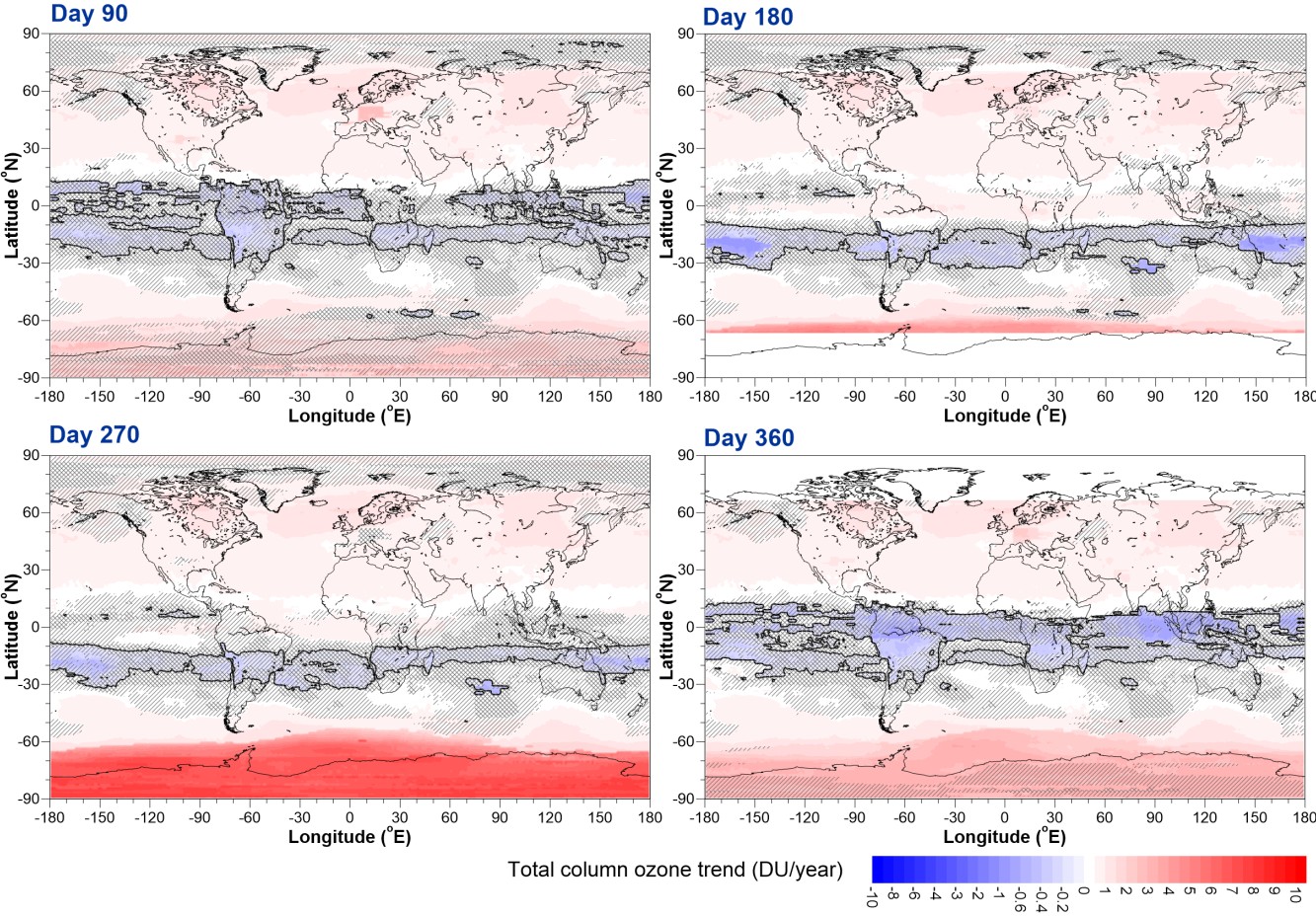

**Figure 17.** The change in TCO trends after 2000 evaluated at (a) day 90, (b) day 180, (c) day 270, and (d) day 360. The solid black line shows zero trend. The same hatching regimen as used in Fig. 16 is used here.

than before 2000) indicate no change in trend before and after 2000. While, since 2000, TCO trends have largely shown a positive change compared to pre-2000 trends (particularly over Antarctica as the ozone hole recovers due to effects of declining stratospheric concentrations of ozone depleting substances), around the middle of the year a large negative change in trend in excess of -1 DU/year is seen in the Southern Hemisphere sub-tropics. The largest of these negative anomalies occurs at 19.5°S,

5   164.375°W and therefore Fig. 18 shows the July mean TCO, regression model fit for July, and the contribution of the offset and trend basis function evaluated at this location. The strong negative change in trend suggested by the regression model is clearly apparent in the observations. This decline in Southern Hemisphere sub-tropical TCO seen here is consistent with other reports of continued declines in tropical ozone after 2000 (Ball et al., 2018). Further research is required into the mechanisms



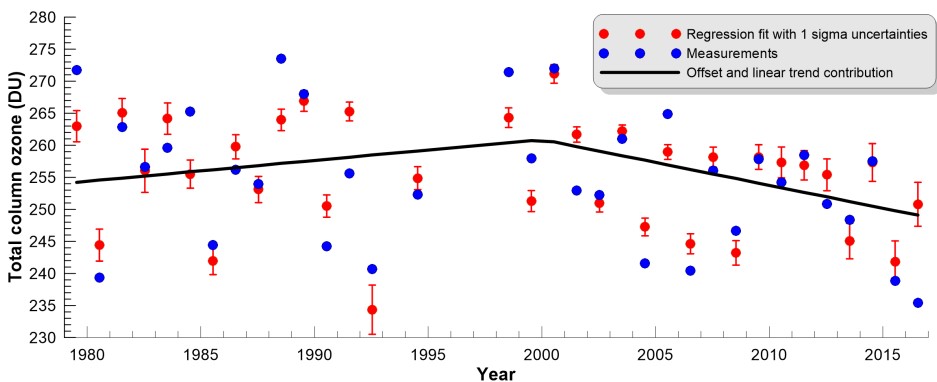

**Figure 18.** The July mean total column ozone at $19.5^\circ$ S, $164.375^\circ$ W (blue dots) together with the regression model values (red dots) and their $1\sigma$ uncertainties, and the contribution to the regression model from the offset and two trend basis functions.

that are driving such decreases in tropical TCO in spite of the effectiveness of the Montreal Protocol and its Amendments and Adjustments in reducing the stratospheric burden of halogens.

## 11 Conclusions

This paper presents the construction of a new version (V3.4) of the NIWA-BS TCO database and the developement of the

5 gap-free BS-filled TCO database. To the extent possible, we have followed the recommendations of GCOS-143 (2010) in establishing a fundamental climate data record for TCO, in particular paying specific attention to tracing all sources of uncertainty through to the final data product. Making the uncertainties available per datum presents a major advancement from the last version of the NIWA-BS TCO database. Comparisons of the NIWA-BS TCO database against the WOUDC database and four independent multi-satellite databases show generally small differences that are within the uncertainties of the NIWA-BS

TCO database. The BS-filled TCO database provides gap-free TCO fields for each day that have been created from a machine-learned algorithm that uses tropopause height and/or potential vorticity at 550 K fields as estimators of the spatial and temporal variability in the TCO fields. Finally, an analysis of trends in unfilled monthly mean TCO fields showed that while many regions of the globe that had been experiencing negative trends in TCO prior to 2000 are now seeing positive TCO trends, there are regions in the tropics (with a bias towards the Southern Hemisphere) where trends since 2000 have become more negative

and over limited regions of the tropics, this decline is statistically significant. The cause of this ongoing negative trend has not been diagnosed here and requires further investigation.



## 12 Data availability

The NIWA-BS TCO database (doi:10.5281/zenodo.1346424, Bodeker et al. (2018)) and the BS-Filled TCO database (doi:10.5281/zenodo.3908787, Bodeker et al. (2020)) are available from http://www.bodekerscientific.com/data/total-column-ozone and from the zenodo archive. Both databases are freely available for non-commercial purposes and are provided as
netCDF files.

*Author contributions.* GEB wrote much of the code for processing the TCO data files into a common format, the code for filling missing data, performed the trend analysis, and wrote the majority of the paper. JN obtained all of the uncertainty information for the different data sets and wrote the code for tracing those uncertainties through the processing chain. JST assisted with the data analysis, the generation of figures, and the writing of the paper. SK ran and debugged the code for applying the corrections to the 17 different TCO data sets and assisted
with the writing of the paper. AS wrote some of the code for homogenising the non-TOMS/OMI data sets. JL wrote much of the C++ code for deriving the statistical correction fields between the non-TOMS/OMI data sets.

*Competing interests.* The authors declare no competing interest.

*Acknowledgements.* We acknowledge the World Meteorological Organization-Global Atmosphere Watch (WMO/GAW) Ozone Monitoring Community, World Ozone and Ultraviolet Radiation Data Centre (WOUDC) for the Dobson and Brewer TCO data retrieved 14 August 2018
from http://woudc.org. A list of all contributing sites is available on https://search.datacite.org/works/10.14287/10000001. We would also like to thank the many satellite teams who provide their data freely to the ozone research community. Without access to these source data, the creation of combined TCO databases such as those reported on here would not be possible. The development of the NIWA-BS TCO database was funded under the New Zealand Deep South National Science Challenge while the development of the BS-filled TCO database was funded internally at Bodeker Scientific. We acknowledge that the isolation that comes with the covid-19 lockdown can also, sometimes, have benefits i.e. it provided the head-space for the lead author to get this paper completed and submitted.



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
