# Peer review of "A Global Total Column Ozone Climate Data Record"

_Earth System Science Data, 2020_

## Referee Comment (RC1) · Anonymous Referee #1 · 3 Nov 2020

This paper presents two global total ozone data sets based on multiple satellite and ground-based data records. The NIWA-BS data set is a update of a 20 years old data set by Bodeker et al., 2001 and NIWA-BS-filled data set as a new set that is based on NIWA-BS, but has extrapolated values in the areas where no NIWA-BS data exist. These data sets are useful for various applications related to the stratospheric ozone. The paper is well written and can be published after minor revisions.

Major comments:

1. From Figure 7, it appears that the NIWA-BS TCO data set has a negative bias against all validation data sets in the Arctic in winter. This should be investigated further. In addition, how large are the differences for the years with large Arctic depletion (e.g., 1997, 2005, 2011)? Does NIWA-BS TCO capture these events correctly? Perhaps such comparisons can be added to Figure 8.

2. Validation of the NIWA-BS-filled data set should be done more thoroughly. Why do not you have a day in March of October when global ozone data with no gaps are available. Then keep only data over the areas where data were available on October 31, 1978 and compare the reconstructed field with the real one. This would confirm the uncertainly estimates for the final reconstructed field. The same could be done for November 1, 1978 data to how the algorithm performs in the case of minor gaps.

3. There should be some validation of the NIWA-BS-filled data set in the polar night areas. They are the most interesting regions. Some total ozone data, such as moon measurements by Dobsons and Bewers as well as from integrated ozonesonde profiles are available.

Specific comments:

p.4, Table 1. Do not use tiny.url. They are shorter, but they may not work in a browser. I've checked the links. Some of them do not work, the other require a password.

p. 5, l. 2 Why is it assumed that the drift is linear? Have you done any tests?

p.5, l 15. Some GB stations (e.g. Mauna Loa) have a bias with satellite data due to high elevation that is not properly accounted by large satellite pixels.

p.5, l 15. Dobson and Brewer instrument have different dependence of stratospheric temperature. This introduces a seasonal difference that could be as high as 2%. Ideally, Dobson data should be adjusted using effective stratospheric temperature.

p.5, l 15 What Dobson and Brewer data were used – all data? DS only? What about SAOZ data? They could be very useful at high latitudes.

P. 16-18. Fig 9-11 are not very informative. Perhaps they should be in a supplement

p. 19, Fig 12. It may be better to show some interesting periods/events rather than "twelve selected months/years"

p. 27, l. 24. Please justify why 2000 was used as the trend turning point.

---

## Referee Comment (RC2) · Anonymous Referee #2 · 6 Jan 2021

The paper presents the datasets of total ozone column (TCO), which are created using the data from several satellite measurements. One dataset is an improved version (v3.4) of NIWA-BS TCO dataset, while another is gap-free BS-filled TCO database. The paper describes the methods used in the construction of the datasets, and some evaluations of dataset, including evaluation of ozone trends.

These datasets are valuable contributions to the ozone research. The paper is well written. Several minor comments and suggestions for paper improvement are below.

MAIN COMMENTS

1. The differences between individual datasets are evaluated using zonal mean values. Does this approach work also in presence of polar vortex? Please add a discussion or

evaluation.

2. P.18: "For some applications, there is a need for gap-free TCO fields". Please indicate these applications.

3. Conservative filling algorithm (Section 9.1). In general, it is easy and advantageous to demonstrate the quality of the filling procedure with artificially created missing data: from a full field some data can be masked and the filling algorithm applied. Then the quality of filling and the quality of uncertainty estimates can be directly evaluated using the true and reconstructed data. For the conservative filling, using the data from previous of following day is a dangerous operation, in my opinion. The air masses are moved, thus such interpolation can result in significant errors, especially in regions of high ozone gradients. The calculation of uncertainties is not described in detail. In particular, it is unclear how the distance from available measurements is taken into account, and this is not seen in Figure 13.

4. Machine learning estimated ozone: It is important to assess the quality of this approach in unexpected ozone conditions, for example Arctic ozone hole in 2011 and 2020. For example, data from 2011 can be excluded from the training dataset, and then tried to reproduce. In general, demonstration of ozone hole evolution in the created datasets would be a very interesting and valuable addition to the paper.

5. Section 10. More details on the regression model would be useful. In particular (a) If possible, repeat the sources of proxies (P.27, L.28) instead of reference to Bodeker et al. (2001), (b) Please add a note on performance of this global fit when data from some months are missing (in polar night conditions). Please add the figure with TCO trends after 2000 (in addition to change trends). This will allow direct comparison with other studies.

SPECIFIC COMMENTS

1) Please write direct links to the datasets, not via tinyurl.com, which do not work
properly.

2) P.4: A map showing locations of Brewer & Dobson stations would be useful. Also a statement on compatibility/similarity of Dobson and Brewer data and their quality would be useful.

3) P.4, L1: Please clarify what you mean by "a higher quality data set": a higher spatial resolution?

4) P.5, around Eq.(2): Since you use integer numbers for f, please indicate also the units of t. It is also worth to note that the choice $N_{f,b}=0$ corresponds to the assumption of a constant drift.

5) P.7, Fig.3 caption: "Regions shaded in grey" –should be "in black"?

6) Table 2. Please indicate that all uncertainties presented in the table are averaged/typical values, which can be different from the uncertainties for a particular location and time.

7) P.9 L 2: should be sigma_Delta ( big Delta)?

8) P.9, Eq.(6), sigma_i should be squared

9) P.10, L.5. Some data are on 1x1 deg grid. Please clarify how the re-gridding is done.

10) Fig.7. It would be useful to indicate also standard deviation of differences.

11) Figures 9 -11: please use smaller color limits.

12) Eq.9, LHS: comma instead of "-"

13) P.23, lines 22-26: The description is not clear, perhaps, an illustration (can be put in the Supplement) would be useful.

---

## Referee Comment (RC3) · Anonymous Referee #3 · 9 Jan 2021

This study improved the long-term total column ozone data record by extending the relevant satellite data input and statistical methods, as well as adding the uncertainty level accompany the released datasets. As an important data source for climate change study, the update of long-term ozone distribution would be helpful for the community to investigate ozone-related topics and analyses. The manuscript could be improved by addressing the following issues:

1) I would suggest to mark the added satellite data (Figure 1) in a way to show the difference with previous data records.

2) P6 L9, what's the 'additional basis function' in particular? Is that simply' set to zero prior to 22 June 2003 and to 1 thereafter'?

3) By spanning the temporal coverage of each of those non-TOMS/OMI data sets, would this implementation introduce for data sets not covering the time span shown in Figure3?

4) P8 L17, what's the setting for atmosphere when simulating the uncertainties using Monte Carlo? Personally, I would like to know what to come up a way to evaluate the uncertainty prorogated from the data sources to the analysis.

5) P10, L5, Would it be possible to further increase the resolution of this product, like length of each side per pixel of 1° or even higher?

6) The polar night shown in Figure 4 is shaded with a different color.

7) I noticed that the latest three years 2013-2016 is not covered in the validation (Table 3), whyïij§

8) Figure 7, The uncertainties seems to be largest around latitudes of 60°, please discuss the reason. In diagram of Dec/Jan/Feb, why the lines go beyond -20 around 60-90°ïij§

9) Is that the areas close to polar night regions are expected to have larger uncertaintiesïij§

10) When using nearest neighbour interpolation to fill missing values, will you test the area of the gap, it might be misleading for filling large areas of missing values.

11) Filling the gap is essentially useful for applications, will the product indicate the regions that is filled with machine learning, while adding uncertainties for these regions might be challenging.

---

## Author Comment (AC1) · 15 May 2021

**Response to referee comments**

May 15, 2021

**1 Anonymous Referee # 1**

This paper presents two global total ozone data sets based on multiple satellite and ground-based data records. The NIWA-BS data set is a update of a 20 years old data set by Bodeker et al., 2001 and NIWA-BS-filled data set as a new set that is based on NIWA-BS, but has extrapolated values in the areas where no NIWA-BS data exist. These data sets are useful for various applications related to the stratospheric ozone. The paper is well written and can be published after minor revisions.

We would like to thank the reviewer for taking the time to review this paper and for their helpful comments which have led to improvements in the paper.

Major comments:

1. From Figure 7, it appears that the NIWA-BS TCO data set has a negative bias against all validation data sets in the Arctic in winter. This should be investigated further.

At the time that we generated this figure, we did investigate this further since the DJF difference in the Arctic did appear anomalous. We found that the TOMS and OMI measurements were generally biased high with respect to the ground-based Dobson and Brewer spectrophotometer measurements over the Arctic in DJF, resulting in a downward correction in the TOMS and OMI measurements at high northern latitudes in this season. Because the corrected TOMS and OMI data then form the basis for the corrections for the other data sets, this induces a downward correction in those data sets. We acknowledge the possibility that the correction required may have been overestimated given that there are only 548 Dobson/Brewer - satellite difference pairs in the Arctic poleward of 60°N. Establishing a better correction for high latitude satellite-based measurements will be a focus for version 3.6 of the NIWA-BS TCO database. Some text has been added to the paper to clarify this.

In addition, how large are the differences for the years with large Arctic depletion (e.g., 1997, 2005, 2011)? Does NIWA-BS TCO capture these events correctly? Perhaps such comparisons can be added to Figure 8.

Recall that the differences are modelled as an offset and trend where the fit coefficients are expanded in Fourier series to capture seasonality and in Legendre polynomials to capture the meridional structure of the differences. Therefore, there is no process for capturing year-to-year variability in the differences. This is appropriate as we don't expect satellite-based measurements to show such variability in any biases with respect to ground-based measurements - they are more likely to show some offset and drift.

We have calculated the area weighted Arctic polar cap mean TCO (poleward of 60°N) for DJF and plotted these in Figure 1 below. In some years it is not possible to calculate a valid DJF mean from the unfilled database due to a paucity of data. The years with low ozone referred to by the reviewer (1997, 2005 and 2011) do indeed show low TCO in both our filled and unfilled databases. Interestingly, 1996 shows a record low; previously, with the unfilled database, a valid value could not be calculated. 2000 also shows very low Arctic polar cap mean TCO with the unfilled value (336 DU) being nearly 25 DU lower than the unfilled value.

2. Validation of the NIWA-BS-filled data set should be done more thoroughly. Why do not you have a day in March or October when global ozone data with no gaps are available. Then keep only data over the areas where data were available on October 31, 1978 and compare the reconstructed field with the real one. This would confirm the uncertainly estimates for the final reconstructed field. The same could be done for November 1, 1978 data to how the

[Figure]

Figure 1: Arctic polar cap area weighted mean TCO for DJF calculated from the filled (red) and unfilled (blue) database. Stars show the year highlighted by the reviewer.

algorithm performs in the case of minor gaps.

We have done many such tests but did not include these in the paper to keep the paper reasonably short (noting that it already has 18 figures). We take the opportunity to provide some additional validation of the filling method in response to this reviewer.

We have taken the original TCO field on day 80 of 1982 (see Figure 2) and removed swaths of data of width 10, 20, 30, 60 and 120° (see Figure 3). The data filling algorithm was then applied to the field shown in Figure 3 to create the fields shown in Figures 4 and 5. The differences between the fields shown in Figure 2 and Figure 4 are shown in Figure 6.

The question now is whether the uncertainties shown in Figure 5 are consistent with the differences shown in Figure 6. We use the formalism of Immler et al. (2010) to quantify the extent to which this is true. Specifically we calculate $k$ as:

$$k = \frac{|m_1 - m_2|}{\sqrt{u_1^2 + u_2^2}} \tag{1}$$

where $m_1$ and $m_2$ are the original and interpolated data values respectively and $u_1$ and $u_2$ are their uncertainties. The field of $k$ values is shown in Figure 7.

There are regions where the uncertainties are under-estimated ($k$ greater than 1.0) and regions where the uncertainties are over-estimated ($k$ smaller than 1.0), but this is to be expected as the uncertainty estimates are, indeed, estimates. The average of all of the $k$ values shown in Figure 7 is 0.892 suggesting that, on average, the uncertainties on the filled values are slightly over-estimated - a value of $k = 1$ denotes that the two values are 'consistent' within their uncertainties (Immler et al., 2010). This is the preferred (conservative) option.

3. There should be some validation of the NIWA-BS-filled data set in the polar night areas. They are the most interesting regions. Some total ozone data, such as moon measurements by Dobsons and Bewers as well as from integrated ozonesonde profiles are available.

[Figure]

Figure 2: The original TCO field on day 80 of 1982.

[Figure]

Figure 3: The TCO field on day 80 of 1982 with swaths of width 10, 20, 30, 60 and 120° having been removed.

[Figure]

Figure 4: The filled TCO field on day 80 of 1982.

[Figure]

Figure 5: The uncertainties on the filled TCO field for day 80 of 1982.

[Figure]

Figure 6: The differences between the original TCO field shown in Figure 2 and the filled field shown in Figure 4.

[Figure]

Figure 7: The $k$ values calculated for the filled field. Only $k$ values for the filled values are shown since they are otherwise everyone 0.0 since there $m_1 = m_2$.

[Figure]

Figure 8: The original TCO field on 21 June 1982. Data north of 60°N were deleted (i.e. poleward of the dashed line).

Rather than trying to source moon measurements from Dobson and Brewer spectrophotometers to conduct the polar night filing validation (noting that these measurements are very sparse both temporally and spatially), we have used a similar method to what was used above to validate the NIWA-BS-filled data set in the polar night areas. Specifically, all data poleward of 60°N where removed over the period 1 June to 15 July from **every** year, noting that the machine-learning algorithm also uses data from neighbouring years to learn how TCO is correlated with 550 K PV and tropopause heights. The deleted data were then infilled using the data filling algorithm. We also note that it is necessary to delete data over a long period within the year otherwise the infilling algorithm simply uses data from day $N-1$ and day $N+1$ to fill the missing data. We select for validation 21 June 1982, i.e. in the middle of the period over which the Arctic data were deleted. The original TCO field is shown in Figure 8, the field with data removed poleward of 60°N is not shown (use your imagination), and the field with data filled poleward of 60°N is shown in Figure 9. In this case the filled TCO values are somewhat overestimated compared to the original data but, on other days, the reverse is true. The question is whether the filled values are consistent with the original data within their uncertainties. Again we calculate the $k$ values as was done above. The map of $k$ values is shown in Figure 10.

The mean of the $k$ values over the Arctic is 1.09 suggesting that the uncertainties calculated on the filled values are realistic in representing the true uncertainty of the filling.

Specific comments:

p.4, Table 1. Do not use tiny.url. They are shorter, but they may not work in a browser. I've checked the links. Some of them do not work, the other require a password.

Thank you for catching this and we have now included direct links and made sure that the data can be found under the links provided. For some data sets, the user will need to create a login and password. The data are freely available but some providers have the requirement to set up an account.

p. 5, l. 2 Why is it assumed that the drift is linear? Have you done any tests?

When we first started this work, around 25 years ago, using TCO measurements from the ground-based Dobson and Brewer spectrophotometer network to correct space-based TCO measurements, we had extensive talks with the Principle Investigators of the space-based instruments (in particular Rich McPeters, PK Bhartia and John Burrows)

[Figure]

Figure 9: The filled TCO field on 21 June 1982. Data poleward of the dotted line (60°N) were filled.

[Figure]

Figure 10: The $k$ values calculated for the Arctic filled data.

to discuss what sort of structure in the drift of the satellite-based measurements we might expect to infer from analyses of temporal structure in the differences between the satellite and ground-based measurements. They strongly advised that we consider only linear drift as any other assumption would likely create spurious structures that were not supported by the difference time series (which are noisy, spatially sparse, and where the spatial coverage varies with time). Only where there were well-documented discontinuities, as with the GOME instrument, did we consider structuring the temporal dependence of the differences (i.e. the drift) as anything other than a linear trend. We did some tests nonetheless and found out that their advice was excellent advice - the difference data did not support any assumed structure in the differences other than a linear drift.

p.5, l 15. Some GB stations (e.g. Mauna Loa) have a bias with satellite data due to high elevation that is not properly accounted by large satellite pixels.

Yes, for the few sites that are at the top of very high and pointy mountains, a good portion of the tropospheric partial ozone column will be lost to the ground-based measurement but will be seen by the satellite (even in overpass mode) which will likely have a much bigger footprint than the peak of the mountain. The corrections that we derive for the satellite-based measurements are largely impervious to this effect since the corrections never rely on any single site. Rather, we fit a 2D surface (as a function of latitude and time, expanded in Legendre polynomials and Fourier series respectively) to the differences calculated at individual sites. The 'stiffness' of this surface, defined by a limited Legendre polynomial expansion, makes the statistically modelled differences field insensitive to such single site effects. That said, in future versions of this database, we will exclude all sites from the difference analyses whose altitudes are above 1 km. We thank the reviewer for bringing this to our attention.

p.5, l 15. Dobson and Brewer instrument have different dependence of stratospheric temperature. This introduces a seasonal difference that could be as high as 2%. Ideally, Dobson data should be adjusted using effective stratospheric temperature.

We are aware of this effect from papers such as Bernhard et al. (2005) and Balis et al. (2007). We are also aware that some recent reprocessing of the Dobson spectrophotometer data has corrected for this effect. Rather than trying to figure out where this correction has and hasn't been applied, and applying it where it hasn't, we have trusted that the TCO data records provided by the World Ozone and UV Data Centre have been processed to be as accurate as possible. As such, we have applied no further corrections to the Dobson spectrophotometer data.

p.5, l 15 What Dobson and Brewer data were used – all data? DS only?

It was stated in the paper that we use direct-sun measurements only. We add that we also used measurements only made with the AD wavelength pair (both ordinary setting and focused images) for the Dobson spectrophotometers, and all data from the Brewer spectrometers.

What about SAOZ data? They could be very useful at high latitudes.

We assessed the volume of SAOZ data that would have been available for such corrections and considered it to be too small to warrant the additional effort to include these data when inferring the satellite-based instrument biases. It was for the same reason that we didn't use DOAS retrievals, microwave radiometer retrievals, or integrated ozonesonde or lidar ozone profiles. But, yes, the availability of SAOZ TCO measurements at high latitudes could be helpful. We will reassess this data source in future constructions of this database. We thank the reviewer for raising this idea.

P. 16-18. Fig 9-11 are not very informative. Perhaps they should be in a supplement

These figures show the spatial and temporal complexity of the structure between our TCO database and the other validation data sets. As such, we believe it is valuable to keep these figures. These figures show in much greater detail the variability in the TCO differences in each month and latitude band; information that is lost when calculating means as in Figure 7. The other two referees felt that these figures had value.

p. 19, Fig 12. It may be better to show some interesting periods/events rather than "twelve selected months/years"

We decided to follow an objective method for which of the many monthly difference fields between the NIWA-BS

and longitudinally resolved ESA CCI database to show. We felt that selectively picking interesting periods/events may only create increased opportunity of perceived selection bias - what may be 'interesting' to us may be completely uninteresting to some other reader who may have, e.g., an interest in the bias in some particular month over some other. We decided, therefore, to show one field for each month of the year and to move sequentially through the years for which difference fields were available. We have updated the figure accordingly.

p. 27, l. 24. Please justify why 2000 was used as the trend turning point.

We have added three sentences to the paper to explain this, i.e. '1999/2000 was prescribed as the trend transition year as this is approximately when stratospheric chlorine and bromine loading peaked (Newman et al., 2007). We also wanted to ensure that the first trend period included data from the late 1990s as there was a greater likelihood of missing data from 1994 to 1998 and we wanted to avoid end-effect-biasing in the calculation of the trends. That said, the conclusions drawn below regarding changes in trend were found to be largely insensitive to the selection of the transition year within 2 years of the selected transition year.'. We thank the reviewer for prompting us to add this detail. It was clearly an omission in the first version of the paper.

**References**

Balis, D., Kroon, M., Koukouli, M.E., Brinksma, E.J., Labow, G., Veefkind, J.P., and McPeters, R.D.: Validation of Ozone Monitoring Instrument total ozone column measurements using Brewer and Dobson spectrophotometer groundbased observations, J. Geophys. Res., 112, D24S46, doi:10.1029/2007JD008796, 2007

Bernhard, G., Evans, R.D., Labow, G.J., and Oltmans, S.J.: Bias in Dobson total ozone measurements at high latitudes due to approximations in calculations of ozone absorption coefficients and air mass, J. Geophys. Res., 110, D10305, doi:10.1029/2004JD005559, 2005

Immler, F.J., Dykema, J., Gardiner, T., Whiteman, D.N., Thorne, P.W., and Vömel, H.: Reference Quality Upper-Air Measurements: guidance for developing GRUAN data products, Atmospheric Measurement Techniques, 3, 1217–1231, doi:10.5194/amt-3-1217-2010, 2010.

---

## Author Comment (AC2) · 15 May 2021

The paper presents the datasets of total ozone column (TCO), which are created using the data from several satellite measurements. One dataset is an improved version (v3.4) of NIWA-BS TCO dataset, while another is gap-free BS-filled TCO database. The paper describes the methods used in the construction of the datasets, and some evaluations of dataset, including evaluation of ozone trends. These datasets are valuable contributions to the ozone research. The paper is well written. Several minor comments and suggestions for paper improvement are below.

We would like to thank the reviewer for taking the time to review this paper and

for their helpful comments which have led to improvements in the paper.

MAIN COMMENTS

1. The differences between individual datasets are evaluated using zonal mean values. Does this approach work also in presence of polar vortex? Please add a discussion or evaluation.

Yes. Other than for the four TOMS and OMI data sets, where the biases are calculated from comparisons between satellite-based and ground-based measurements at individual sites (but them modelled without zonal structure), corrections for the other data sets are determined through comparisons of zonal means against the corrected TOMS and OMI data. If the zonal sampling of a satellite was biased, e.g., if the satellite only made measurements between say $0°E$ and $90°E$, then its zonal mean could be very biased with respect to the zonal mean from an unbiased satellite-based instrument, especially in the presence of zonal asymmetries as occurs when the polar vortex is pushed off the pole, e.g. by a large planetary wave of zonal wavenumber 1, as is pointed out by the reviewer. But this is seldom, if ever, the case, i.e. the TCO measurements from the satellites we have used are zonally unbiased and the comparisons do not suffer from such effects. Cognisant that this may have not been the case for the more spatially sparse SBUV measurements, we did include an additional source of uncertainty in the calculation of SBUV zonal means, as discussed on pages 9 and 10 of the original paper. We have added material towards the top of Section 4 to this effect, i.e. 'There could be a danger here that in the case of biased longitudinal sampling by one satellite compared to another, that the zonal means would be biased but without these differences arising from any intrinsic biases between the satellite-based measurements. Only the SBUV measurements were sparse and corrections for this potential sampling bias were derived as discussed below.'

2. P.18: "For some applications, there is a need for gap-free TCO fields". Please indicate these applications.

We have now expanded this sentence to say 'For some applications, there is a need for gap-free TCO fields, e.g., TCO fields for validating chemistry-climate models which generate TCO fields over the entire globe for each day of the year.'

3. Conservative filling algorithm (Section 9.1). In general, it is easy and advantageous to demonstrate the quality of the filling procedure with artificially created missing data: from a full field some data can be masked and the filling algorithm applied. Then the quality of filling and the quality of uncertainty estimates can be directly evaluated using the true and reconstructed data.

This is a good idea and the same suggestion was made by Referee 1. The results of exactly that sort of evaluation (an additional 10 figures), are provided in the response to Referee 1. We decided not to include all of this additional validation of the filling method in the paper as (1) the paper already has 18 figures and adding another 10 would have been untenable, and (2) because the response to the referee comments accompany the paper, we felt that readers who were interested in a deeper validation of the filling technique could refer to our response to Referee 1.

For the conservative filling, using the data from previous of following day is a dangerous operation, in my opinion. The air masses are moved, thus such interpolation can result in significant errors, especially in regions of high ozone gradients.

We never just use the value from the previous or following day. As stated in the paper, we linearly interpolate between values on day $N + 1$ and day $N - 1$ to estimate a value for day $N$. We have found this to be very robust and as can be validated by inspecting a TCO time series for any location on the global. Given the very long

photochemical lifetime of TCO, the time series tend to be rather smooth suggesting that linear interpolation between neighbouring days is an acceptable way to estimate missing data. The assumption that is made is not that ozone is unchanged from one day to the next, but that TCO changes approximately linearly from one day to the next. Essentially this is equivalent to a CFL (Courant–Friedrichs–Lewy) condition for TCO.

The calculation of uncertainties is not described in detail. In particular, it is unclear how the distance from available measurements is taken into account, and this is not seen in Figure 13.

Yes, we have not described the calculation of the uncertainties in the longitudinal linear interpolation in great detail. We were trying to keep the paper as succinct as possible. Calculating the uncertainties on such longitudinally interpolated values is non-trivial. The challenges of estimating the uncertainties on such interpolations are described nicely in Fassò et al. (2020). We have used a rather simple method for estimating the uncertainty on the longitudinal interpolation, as follows:

$$
\begin{aligned}
\alpha &= (\phi - \phi_1)(\phi_2 - \phi_1) \\
T_1 &= (1 - \alpha) \times \sigma_1^2 \\
T_2 &= \alpha \times \sigma_2^2 \\
\sigma_{interp} &= \sqrt{T_1 + T_2}
\end{aligned}
\tag{1}
$$

where $\phi$ is the longitude at which the interpolation is being performed, $\phi_1$ is the longitude to the west where there is a non-null value, $\phi_2$ is the longitude to the east where there is a non-null value, $\sigma_1$ is the uncertainty on the non-null value to the west and $\sigma_2$ is the uncertainty on the non-null value to the east.

The reason that this inflation of uncertainties in conducting the longitudinal interpolation is not apparent in Figure 13, is that much of the interpolation in Figure 13 is through interpolation between day $N - 1$ and day $N + 1$ which takes precedent over

the longitudinal interpolation.

4. Machine learning estimated ozone: It is important to assess the quality of this approach in unexpected ozone conditions, for example Arctic ozone hole in 2011 and 2020. For example, data from 2011 can be excluded from the training dataset, and then tried to reproduce. In general, demonstration of ozone hole evolution in the created datasets would be a very interesting and valuable addition to the paper.

We agree and this has been done in some detail in response to Reviewer 1. We have not explicitly done this for 2011 and 2020. In suggesting that some validation can be achieved by withholding, e.g., 2011, from the data set, we believe that the reviewer is seeing this as a neural network-based solution. It is not. Rather, it is a regression approach where the regression model establishes a local (in space and time, where neighbouring years are included in the time window) statistical dependence of TCO on local tropopause height and local potential vorticity at the 550 K surface. We could train the regression model to generate TCO fields for all of 2011 when all TCO fields from 2011 are excluded, but this would not be an appropriate validation since the regression model has been constructed to assume that there will be TCO fields within a few days of the target field which is always the case. We have not done as the reviewer has suggested for 2020 since our database terminates in 2016.

5. Section 10. More details on the regression model would be useful. In particular (a) If possible, repeat the sources of proxies (P.27, L.28) instead of reference to Bodeker et al. (2001),

We did not include details of the El Niño Southern Oscillation (ENSO), solar cycle, Mt. Pinatubo and El Chichón basis functions in this paper as they have been published in Bodeker et al. 2001b and Bodeker et al. 1998, and both papers are publicly available. Including the additional information here is unnecessarily verbose

and would lengthen the already long paper even further.

(b) Please add a note on performance of this global fit when data from some months are missing (in polar night conditions).

It is not clear to us what the reviewer is referring to here as the fit is not global, i.e., equation (18) is applied completely independently at each latitude and longitude. As to the challenges of obtaining trends during the polar night, while the Fourier expansion in the regression model fit coefficient (accounting for trend structure at annual, 6-monthly, and 4-monthly periods) is somewhat robust against data in missing months, there is a danger of over-fitting at high latitudes in winter and obtaining unreliable trends in regions of polar darkness. This is why we exclude trends in these regions in Figures 16 and 17.

Please add the figure with TCO trends after 2000 (in addition to change trends). This will allow direct comparison with other studies.

Yes, this is a good idea and we have added this figure as the new Figure 18.

SPECIFIC COMMENTS

1) Please write direct links to the datasets, not via tinyurl.com, which do not work properly.

Thank you for catching this and we have now included direct links and ensured that the data can be found under the links provided.

2) P.4: A map showing locations of Brewer & Dobson stations would be useful.

We have decided not to do this for two reasons:

1. It would add yet another figure to the paper.

2. As a nice interactive map of all WOUDC sites is provided at https://woudc.org/data/stations/index.php.

We felt that an additional figure of all the measurement sites is unwarranted and any reader will find the information (easily available) on the well maintained and updated WOUDC website. We have added a pointer to the this URL in the paper towards the beginning of Section 3.

2)Also a statement on compatibility/similarity of Dobson and Brewer data and their quality would be useful.

We have added the following sentence to the paper 'We assumed in all cases that the Dobson and Brewer spectrophotometer data submitted to the WOUDC were the highest quality data available, that all possible corrections to improve the quality of the data had been made prior to submission, and that the measurements from these two networks were unbiased with respect to each other. Assumed uncertainties on the measurements from these two networks are presented below'. We felt in unnecessary to provide a more detailed discussion of the uniformity of the data either within or between these two measurement networks as many other papers have conducted such analyses, e.g. Bojkov et al. (1986), Bojkov et al. (1990), Zerefos et al. (1992), Bokjov et al. (1995).

3) P.4, L1: Please clarify what you mean by "a higher quality data set": a higher spatial resolution?

We have replaced this sentence with 'While TOMS and OMI data are provided in gridded data files, the original overpass data, unlike the gridded data set where several measurements from different times through the day might be averaged within a grid cell, provide location-specific measurements that are more suitable for comparison with the ground-based measurement networks, i.e. the overpass measurements are specific to a latitude, longitude and date/time.' to be more explicit in what we had previously meant when we had only said 'a higher quality data set'.

4) P.5, around Eq.(2): Since you use integer numbers for f, please indicate also the units of t. It is also worth to note that the choice Nf,b=0 corresponds to the assumption of a constant drift.

We have added just after equation (1) that time is measured in decimal years. The choice of $N_{F,\beta} = 0$ is not that the drift is constant but that the drift is seasonally independent. A sentence has been added to the paper to clarify this.

5) P.7, Fig.3 caption: "Regions shaded in grey" –should be "in black"?

Thank you and we have corrected this error.

6) Table 2. Please indicate that all uncertainties presented in the table are averaged/typical values, which can be different from the uncertainties for a particular location and time.

As suggested by the reviewer, we have amended the table caption to 'Typical uncertainties on the source data sets as reported in the references provided and as used in the construction of the TCO databases. Note that the uncertainty on any particular measurement may differ from the typical values quoted in this table'. We have also removed the links from Table 2 and included the links as references.

7) P.9 L 2: should be sigma_Delta (big Delta)?

The reviewer is correct and we have changed the small delta to a big delta.

8) P.9, Eq.(6), sigma_i should be squared

Thank you for spotting this mistake, which we have now corrected.

9) P.10, L.5. Some data are on 1x1 deg grid. Please clarify how the re-gridding is done.

Where source data were provided at $1° \times 1°$ resolution, bilinear interpolation was used to resample the data to $1.25° \times 1°$ resolution. A sentence to this effect has been added early in Section 2.

10) Fig.7. It would be useful to indicate also standard deviation of differences.

We agree and have made a new version of Figure 7.

11) Figures 9 -11: please use smaller color limits.

If by 'use smaller color limits' the reviewer means that we should reduce the colour range on each plot, we prefer not to for two reasons: (1) We would like to capture the extreme values and (2) we would like to use the same color range on each figure to make them comparable.

12) Eq.9, LHS: comma instead of "-"

This has been corrected.

13) P.23, lines 22-26: The description is not clear, perhaps, an illustration (can be put in the Supplement) would be useful.

This paper is not accompanying by a Supplement and we were reluctant to now add a Supplement just to accommodate this single additional figure. Rather than adding a figure to the paper, we have described in greater detail how the $A_{proxy}$ values are calculated.

**References**

Balis, D., Kroon, M., Koukouli, M.E., Brinksma, E.J., Labow, G., Veefkind, J.P., and McPeters, R.D.: Validation of Ozone Monitoring Instrument total ozone column measurements using Brewer and Dobson spectrophotometer groundbased observations, J. Geophys. Res., 112, D24S46, doi:10.1029/2007JD008796, 2007

Bojkov, R.D., Bishop, L., and Fioletov, V.E.: Total ozone trends from quality-controlled ground-based data (1964-1994), J. Geophys. Res., 100D12, 25867-25876, 1995

Bojkov, R., Bishop, L., Hill, W.J., Reinsel, G.C., and Tiao, G.C.: A statistical trend analysis of revised Dobson total ozone data over the Northern Hemisphere, J. Geophys. Res., 95, D7, 9785-9807, 1990

Bojkov, R.D.: The 1979-1985 ozone decline in the Antarctic as relected in ground based observations, Geophys. Res. Lett., 13, 12, 1236-1239, 1986

Fassò, A., Sommer, M., and von Rohden, C.: Interpolation uncertainty of atmospheric temperature radiosoundings, Atmospheric Measurement Techniques, doi:10.5194/amt-2020-161, 2020

Zerefos, C.S., Bais, A.F., Ziomas, I.C., and Bojkov, R.D.: On the relative importance of quasi-biennial oscillation and El Nino southern oscillation in the revised Dobson total ozone records, J. Geophys. Res., 97, D9, 10135-10144, 1992

---

## Author Comment (AC3) · 15 May 2021

This study improved the long-term total column ozone data record by extending the relevant satellite data input and statistical methods, as well as adding the uncertainty level accompany the released datasets. As an important data source for climate change study, the update of long-term ozone distribution would be helpful for the community to investigate ozone-related topics and analyses. The manuscript could be improved by addressing the following issues:

We would like to thank the reviewer for taking the time to review this paper and for their helpful comments which have led to improvements in the paper.

1) I would suggest to mark the added satellite data (Figure 1) in a way to show the difference with previous data records.

We have now done this and have added explanatory text to the figure caption. Newly added satellite are shown with an additional dashed line.

2) P6 L9, what's the 'additional basis function' in particular? Is that simply' set to zero prior to 22 June 2003 and to 1 thereafter'?

Yes, as stated in the manuscript. This basis function then allows for an additional offset across the 22 June 2003 transition.

3) By spanning the temporal coverage of each of those non-TOMS/OMI data sets, would this implementation introduce for data sets not covering the time span shown in Figure 3?

We do not understand the referee's comment. As shown in Figure 1, the TOMS+OMI record spans the full period of the data set except for a gap of approximately 1 year in 1995. As such, the corrected (biases and offsets removed), TOMS+OMI data set provides a valid standard against which all other satellites can be compared.

4) P8 L17, what's the setting for atmosphere when simulating the uncertainties using Monte Carlo? Personally, I would like to know what to come up a way to evaluate the uncertainty prorogated from the data sources to the analysis.

We don't understand what the referee is referring to by "what's the setting for atmosphere". Essentially we need to calculate uncertainties on the 2D manifold (latitude and time) fitted to the difference pairs and propagate those uncertainties through

to the final data product, i.e. the derived corrections are themselves uncertain and when applying those corrections an additional uncertainty is introduced which must be included into the uncertainty budget. We do this by generating different, but statistically equivalent, versions of the difference data set by Monte Carlo sampling the uncertainties on the differences and adding those to the original differences. We create 100 such manifolds. The mean and standard deviation of the 100 manifolds provides the final difference field and its uncertainty.

5) P10, L5, Would it be possible to further increase the resolution of this product, like length of each side per pixel of 1° or even higher?

No. Essentially we are constrained by the coarsest resolution of the source data sets which is $1.25° \times 1.0°$ for the TOMS data.

6) The polar night shown in Figure 4 is shaded with a different color.

True. This is stated in the figure caption.

7) I noticed that the latest three years 2013-2016 is not covered in the validation (Table 3), why?

Thanks for catching this and we have made the necessary corrections to Table 3.

8) Figure 7, The uncertainties seems to be largest around latitudes of 60°, please discuss the reason.

We have not diagnosed why the uncertainties on the NIWA-BS TCO values maximize around 60° latitude. That's simply what emerged from tracing the various sources of

uncertainty through to the final data product.

In diagram of Dec/Jan/Feb, why the lines go beyond -20 around 60-90°?

A detailed discussion of the large differences between the NIWA-BS TCO database and the other validation databases is given in response to comment 1 of Referee 1. We refer the referee to that response.

9) Is that the areas close to polar night regions are expected to have larger uncertainties?

While we are not entirely sure what the referee is referring to here, we can state that because TCO values tend to be higher at higher latitudes, and that we have assumed fixed percentage uncertainties on many of the source data sets, that the absolute uncertainties tend to maximize at higher latitudes. For the filled version of the database it is certainly true that the uncertainties maximize over the polar night regions since this is where the ML-based filling occurs and these can be accompanied by large uncertainties.

10) When using nearest neighbour interpolation to fill missing values, will you test the area of the gap, it might be misleading for filling large areas of missing values.

Yes, as stated in the paper, for the nearest neighbour interpolation, only single cells neighboured by non-null cells are filled.

11) Filling the gap is essentially useful for applications, will the product indicate the regions that is filled with machine learning, while adding uncertainties for these regions might be challenging.

There are two ways in which users of the database can detect where filling has occurred:

1. Examine both the unfilled and filled database and see where the latter has values and the former does not.

2. Examine the uncertainties on the filled TCO fields. Uncertainties tend to be much higher where values have been filled.

We agree with the reviewer that deriving uncertainties for the filled values is challenging. A validation of the uncertainties on ML-filled regions is provided in response to comment 3 of Referee 1. We refer this referee to that response.